

# A global hydrological simulation to specify the sources of water used by humans

Naota Hanasaki[12], Sayaka Yoshikawa[3], Yadu Pokhrel[4], Shinjiro Kanae[3]

5   [1] National Institute for Environmental Studies, Tsukuba, Japan
    [2] International Institute for Applied System Analyses, Laxenburg, Austria
    [3] Department of Civil and Environmental Engineering, Tokyo Institute of Technology, Tokyo, Japan
    [4] Department of Civil and Environmental Engineering, Michigan State University, East Lansing, USA

*Correspondence to*: Naota Hanasaki (hanasaki@nies.go.jp)

**Abstract.** Humans abstract water from various sources to sustain their livelihood and society. Some global hydrological models (GHMs) include explicit schemes of human water abstraction, but the representation and performance of these schemes remain limited. We substantially enhanced the water abstraction schemes of the H08 GHM. This enabled us to estimate water abstraction from six major water sources, namely, river flow regulated by global reservoirs (i.e., reservoirs regulating the flow

of the world's major rivers), aqueduct water transfer, local reservoirs, seawater desalination, renewable groundwater, and nonrenewable groundwater. In its standard setup, the model covers the whole globe at a spatial resolution of $0.5° \times 0.5°$, and the calculation interval is one day. All the interactions were simulated in a single computer program and the water balance was always strictly closed at any place and time during the simulation period. A global hydrological simulation was conducted to validate the performance of the model for the period of 1979-2013. The simulated water fluxes for water abstraction were

validated against those reported in earlier publications, and showed a reasonable agreement at the global and country level. The simulated monthly river discharge and terrestrial water storage (TWS) for six of the world's most significantly human-affected river basins were compared with river gauging observations and a satellite product of the Gravity Retrieval and Climate Experiment (GRACE) mission. It showed that the simulation including the newly added schemes outperformed the simulation without them. The results indicated that, in 2000, of the 3628 $km^3yr^{-1}$ global freshwater requirement, 2839 $km^3yr^{-1}$

was taken from surface water and 789 $km^3yr^{-1}$ from groundwater. Streamflow, aqueduct water transfer, local reservoirs, and seawater desalination accounted for 1786, 199, 106, and 1.8 $km^3yr^{-1}$ of the surface water, respectively. The remaining 747 $km^3yr^{-1}$ freshwater requirement was unmet, or surface water was not available when and where it was needed in our simulation. Renewable and nonrenewable groundwater accounted for 607 and 182 $km^3yr^{-1}$ of the groundwater total, respectively. Each source differed in its renewability, economic costs for development, and environmental consequences of usage. The model is

useful for performing global water resource assessments by considering the aspects of sustainability, economy, and environment.



## 1. Introduction

Water is an indispensable resource for human society. The securing of water resources is an important global challenge in the 21$^{st}$ century, because the demand for water is projected to increase due to the growing population, increased economic activity, and changing climate (Oki and Kanae, 2006). To quantify global water availability and use in the past, present, and future, a

number of global hydrological models (GHMs) have been developed to provide an explicit representation of human water use, including H08 (Hanasaki et al., 2008a,b, 2010), WaterGAP (Alcamo et al., 2003; Döll et al., 2003, 2012, 2014), LPJmL (Gerten et al., 2004; Rost et al., 2008, Biemans et al., 2011), PCR-GLOBWB (van Beek et al., 2011; Wada et al., 2011, 2014), WBMplus (Vörösmarty et al., 1989; Wisser et al. 2010), HiGW-MAT (Pokhrel et al., 2012a,b, 2015), and others. The history of model development is well summarized in Nazemi and Wheater (2015a,b), Bierkens (2015), Sood and Smakhtin (2015),

and Pokhrel et al. (2016).

The fundamental objectives of GHMs are twofold. The first objective is to estimate flows and stocks of natural hydrological components (e.g., river water, soil moisture, and groundwater) and human water use at sufficiently high spatial and temporal resolution. This objective has been largely achieved in the last two decades by developing physical hydrological models to solve the surface water balance (e.g., Döll et al., 2003; Gerten et al., 2004; Hanasaki et al., 2008a; van Beek et al., 2011),

developing water use models to estimate irrigation, industrial, municipal, and other water requirements (e.g., Döll and Siebert, 2002; Alcamo et al., 2003; Rost et al., 2008; Hanasaki et al., 2008b, Wada et al., 2011; Flörke et al., 2013), and developing global gridded data to provide the boundary conditions of such models (e.g., Döll et al., 2003; Siebert et al., 2005; Lehner et al., 2011). The second objective is to represent the interaction between natural hydrology and human water use within a single modeling framework. Water abstraction from rivers was first implemented in GHMs (e.g., Hanasaki et al., 2008b; Rost et al.,

2008). This enabled the GHMs to represent the fundamental nature-human interactions, in which upstream water abstraction reduces water availability in downstream areas.

One of the remaining challenges of GHMs is to enable water abstraction from various water sources including the effects of water infrastructure. Water sources can be separated into surface water and groundwater. Groundwater is a renewable water source, but it could be depleted if the volume of water abstraction exceeds the recharge (e.g., Wada et al., 2010). Hence it

should be further separated into renewable and nonrenewable (overexploited) categories. River flow has been the dominant surface water source for humans since ancient times. Because river flow has substantial temporal variations, it is regulated and stored in artificial reservoirs and ponds, which can be used in periods of low flow. Furthermore, because river flow is accessible to regions located along the channel, aqueducts have been constructed to transfer it to regions located further away. This infrastructure has a critically important role in enhancing the utility of river flow. Other water sources include lakes, glaciers,

and seawater desalination. Seawater desalination is an emerging water source in arid coastal regions and has been boosted by recent technological advances (Ghaffour et al., 2013). Most advanced GHMs have already implemented some of the water sources referred to above (see Table S1), but none include all of them in a single hydrological model.





To overcome this limitation, we enhanced the H08 model. The H08 model is one of the earliest models to provide global simulations by considering interactions between the natural water cycle and human water use. The human activities considered in the model include water abstraction for irrigation water, operation of reservoirs, and water abstraction from rivers (see Appendix A for technical details). Our enhancement enabled H08 to: (1) explicitly represent groundwater recharge and

availability, (2) estimate the geographical distribution and volume of water abstracted from groundwater, while distinguishing between the renewable and nonrenewable portions, (3) estimate water transferred over a distance via aqueducts, (4) better represent local reservoirs and estimate water abstraction from them, (5) estimate seawater desalination, and (6) better represent the process of water abstraction and estimate delivery loss and return flow. By incorporating these six schemes, H08 has become one of the most detailed GHMs for attributing the water sources available to humanity (see Table S1).

## 2.    Methods

### 2.1. Newly added schemes

Six schemes or additional components were developed and implemented into H08, namely, groundwater recharge, groundwater abstraction, aqueduct water transfer, local reservoirs, seawater desalination, and return flow and delivery loss schemes. All six schemes were added to the original H08, with the exception of the local reservoir scheme, which was replaced

with that of the original H08 (Hanasaki et al. 2008a,b, 2010, 2013a,b). Figure 1 shows a schematic diagram of the enhanced H08.

In this subsection, each scheme is described individually. Each description begins with a brief technical review of existing schemes, and is followed by detailed model formulations. Other than the newly added schemes, the formulations of H08 were identical to those of the original H08, which are reported in Hanasaki et al. (2008a,b, 2010). See Appendix A1 for concise

model descriptions.

### 2.1.1.    Groundwater recharge

Groundwater flow is a fundamental hydrological process. Although it is difficult to represent the groundwater process precisely at any spatial scale in hydrological modeling (e.g., Healey, 2010), let alone at the global scale, considerable efforts have been made in recent decades. Döll et al. (2002) and Döll and Fiedler (2008) first developed a model to estimate groundwater recharge

globally and incorporated it into the WaterGAP model. They estimated the fraction of total runoff that recharges aquifers by using the available global digital maps of slope, soil texture, geology, and permafrost. The approach is simple and computationally inexpensive, but the results are largely dependent on various parameters. An alternative approach is to estimate groundwater recharge by solving the Richards' equation (Richards, 1931). This is suitable for hydrological models with multiple soil moisture layers and an explicit physical representation of the soil moisture dynamics (e.g. Fan et al., 2007,

Niu et al., 2007). Some studies have taken an intermediate position between these approaches. Van Beek and Bierkens (2008) developed a GHM that included a linear groundwater reservoir and incorporated it into the PCRaster Global Water Balance





(PCR-GLOBWB) model. Koirala et al. (2014) developed a groundwater sub-model for the Human Intervention and Ground Water coupled MATSIRO (HiGW-MAT) model by adopting the statistical-dynamical approach of Yeh and Eltahir (2005).

To represent groundwater recharge, the algorithm developed by Döll and Fiedler (2008) was adopted and added to the land surface hydrology sub model of H08. Their approach was compatible with H08, which has only one soil layer and is considered reliable because their model has been validated in numerous subsequent publications (e.g., Döll et al., 2012, Döll et al., 2014). A complete description is provided in the appendixes of Döll and Fiedler (2008), but the methodology is briefly described here. Groundwater recharge ($Qrc$ [kg m$^{-2}$ s$^{-1}$]) is formulated as below:

$$Qrc = min(Qrc_{max}, f_r \cdot f_t \cdot f_h \cdot f_{pg} \cdot Qtot) \tag{1}$$

where $Qrc_{max}$ is the maximum groundwater recharge [kg m$^{-2}$ s$^{-1}$], $f_r$ is a relief-related factor ($0 < f_r < 1$), $f_t$ is a soil-texture-related factor ($0 < f_t < 1$), $f_h$ is a hydrogeology-related factor ($0 < f_h < 1$), $f_{pg}$ is a permafrost/glacier-related factor ($0 < f_{pg} < 1$), and $Qtot$ is the total runoff [kg m$^{-2}$ s$^{-1}$]. $Qrc_{max}, f_r, f_t, f_h$, and $f_{pg}$ are determined by the look-up-tables provided in Tables A1-A4 of Döll and Fiedler et al. (2008), which link these factors with global geographical maps.

Four global maps were used as the inputs of the scheme. The maps used in this study differed from those used in Döll and Fiedler (2008) because the maps they referred to have been substantially updated. For relief, the Global Relief Data were used, which are included in the Harmonized World Soil Database v 1.1 (HWSD; FAO et al., 2012). The data provide the global distribution of relief in eight categories. For soil texture, the Soil Texture Map for the Global Soil Wetness Project Phase 3 (GSWP3; http://hydro.iis.u-tokyo.ac.jp/~sujan/research/gswp3/soil-texture-map.html) was used. The soil texture data were subdivided into 13 classes covering the whole globe at the spatial resolution of 0.5° × 0.5°. For hydrogeological data, OneGeology (http://www.onegeology.org/) was used, which is an international initiative of the world's various geological surveys. For permafrost and glacier data, the Circum-Arctic Map of Permafrost and Ground-Ice Conditions by the National Snow Ice Data Center of the USA was used (Brown et al., 2002).

The groundwater recharge drains into the renewable groundwater reservoir (see Fig 1). The water balance of the renewable groundwater reservoir is expressed as:

$$\frac{dSrgw}{dt} = Qrc - Qb - \frac{WArgw}{A} \tag{2}$$

where $Srgw$ is the storage of the renewable groundwater reservoir [kg m$^{-2}$], $Qb$ is the baseflow [kg m$^{-2}$ s$^{-1}$] $WArgw$ is the total withdrawal-based abstraction from renewable groundwater [kg s$^{-1}$], and A is the area of a grid cell [m$^2$]. Importantly, there is no capillary rise (i.e., water in the renewable groundwater reservoir does not move into the soil moisture reservoir). The baseflow ($Qb$) [kg m$^{-2}$ s$^{-1}$] or outflow from the renewable groundwater reservoir is estimated as:

$$Qb = \frac{Srgw_{max}}{\tau}\left(\frac{Srgw}{Srgw_{max}}\right)^{\gamma} \tag{3}$$

where $Srgw_{max}$ is the maximum storage capacity of the renewable groundwater reservoir [kg m$^{-2}$], $\tau$ is a time constant [s], and $\gamma$ is a shape parameter [-]. In this study, we set $Srgw_{max}, \tau$, and $\gamma$ at 150 kg m$^{-2}$, 100 days, and 2.0, respectively. These numbers were empirically derived. For $\tau$, Döll et al. (2012) also adopted the same number. Equations (2-3) were solved explicitly, or the fluxes were determined by the state variables of the previous time step.



In addition to the renewable groundwater, we added a nonrenewable groundwater reservoir (see Fig. 1). This is a hypothetical groundwater reservoir that stores a limitless volume of water, and is isolated from both soil moisture and the renewable groundwater reservoir (i.e., no recharge and no capillary rise), which is explained in the next subsection.

### 2.1.2. Groundwater abstraction

Groundwater is an essential source of water for humans, and accounts for 26% of the total water withdrawal in 2010 (Margat and van der Gun, 2013). Until recently, some GHMs that included groundwater reservoirs explicitly incorporated groundwater abstraction. To the best of our knowledge, Wada et al. (2010) was the first study to combine the modeled groundwater recharge and statistics-based abstraction. The authors spatially distributed national groundwater use statistics and calculated the balance of groundwater recharge and abstraction. Subsequently, Döll et al. (2012; 2014), Wada et al. (2014), and Pokhrel et al. (2015)
improved their models to better represent groundwater abstraction. The algorithm for groundwater abstraction typically consists of two parts. The first separates the groundwater abstraction requirement from the total water requirement, and the second fulfills the groundwater requirement from groundwater resources. For the first part that separates the groundwater requirement, the earlier studies can be roughly classified into two types. One relies on national statistics of groundwater usage (e.g., Döll et al., 2012) and the other on conceptual models (e.g., Wada et al., 2014). The former has the advantage of
constraining the results by statistics, but it becomes problematic when the model is applied to regions and periods where data are lacking. The latter has the opposite strengths and weaknesses. For the second part, regarding the fulfillment of the groundwater requirement, some models take water from the groundwater reservoir (e.g., Döll et al., 2012), while others take water from the baseflow (e.g., Wada et al., 2014).

To represent groundwater abstraction, an algorithm similar to Döll et al. (2012) was added to the water abstraction sub model
of H08. As mentioned earlier, there is a statistical approach (e.g., Döll et al., 2012) and a modeling approach (Wada et al., 2014) for separating the groundwater requirement from the total water requirement. We tested both and found a substantial difference between the two approaches (data not shown). We finally adopted the former method because it had less uncertainty in reproducing the historical past.

Similarly to Döll et al. (2012), we estimated the fractional contribution of surface and groundwater abstraction toward the total
water requirement for each sector. For irrigation, we estimated the surface and groundwater fraction from the area of irrigated cropland. The global distribution of the area equipped with and without groundwater irrigation facilities was provided by Siebert et al. (2010), and covers the world at a spatial resolution of 0.5° × 0.5°. We assumed that all irrigation water was supplied by groundwater (surface water) if cropland was equipped with (without) groundwater irrigation facilities. To estimate the fractional contribution of surface and groundwater for industrial and municipal purposes, the International Groundwater
Resources Assessment Centre (IGRAC, 2004) groundwater use database was used. This provides sector- and source-specific (surface and groundwater) water use for nations from 1995. For the nations where both sector and source information was available the fraction was used throughout the simulation period. For the nations where data were lacking (i.e., the majority of nations), we used the fraction for representative countries in the region, as shown in Table A1. We adopted the Food and



Agriculture Organization of the United Nations (FAO) regional classification, which subdivides the world into 22 regions. For each region, representative countries were selected for which the complete data were available. If data were available for more than one country, the country with the larger population was used. The fractional contribution of the overall water requirement assigned to groundwater is shown in Fig. 2.

Once the daily water requirement was assigned to groundwater in each grid cell, water was first abstracted from the renewable groundwater reservoir to meet the overall requirement. If the renewable groundwater was depleted, water was abstracted from the nonrenewable groundwater reservoir, which corresponds to the overexploitation of groundwater in reality. In a mathematical form, the withdrawal-based water abstraction (i.e., including return flow and delivery loss) from renewal and nonrenewal groundwater ($Wrgw$ and $Wngw$) [kg s$^{-1}$] is expressed as follows:

$$Wrgw = min(f_{gw} \cdot Qreq, Srgw/\Delta t) \tag{4}$$

$$Wngw = f_{gw} \cdot Qreq - Wrgw \tag{5}$$

$$Wgw = Wrgw + Wngw \tag{6}$$

where $Qreq$ is the total water requirement [kg s$^{-1}$] and $f_{gw}$ is the fraction of the overall water requirement assigned to groundwater [-]. $\Delta t$ denotes the calculation interval [s].

**2.1.3.  Aqueduct water transfer**

River water is in some cases transferred over long distances through aqueducts (i.e., canals, pipes, and others). For example, the Colorado River Aqueduct in the USA extends for nearly 400 km, transferring the flow of the Colorado River to southern California. Several GHMs have incorporated hypothetical algorithms to express this transfer, but they are simplistic. The WaterGAP model allows water to be taken from the neighboring cell with the largest upstream area (Döll et al., 2012). This is

a highly conceptual, but practical, approach because information regarding aqueducts is not available for all regions worldwide. We modeled water transfer via aqueducts as described below. A schematic diagram is shown in Fig. 3. First we defined an explicit and implicit aqueduct. An explicit aqueduct was an individual aqueduct whose existence could be confirmed from the literature, while for an implicit aqueduct there was a general inference that major rivers would supply water to the cells nearby if necessary, regardless of the confirmed existence of an aqueduct.

For both types of aqueduct, global digital maps were prepared. For explicit aqueducts, the geographical locations of 55 major aqueducts were identified. We collected publications relating to major aqueducts in six countries, namely, China, Egypt, India, Israel, Pakistan, and the USA (listed in Table A2) and compiled them electronically on geographic information system (GIS) software. We selected the aqueducts longer than 50 km or the length of the edge of a $0.5° \times 0.5°$ grid cell. Then the origin (i.e., the point at which an aqueduct is diverted from the river), destination and route of each aqueduct were georeferenced on the

digital river network. The maps for the western USA, central China, the eastern Mediterranean region, and the Indian Subcontinent are shown in Fig. 4. Because we set the base year at 2000, the south-north water transfer in China, which is facilitated by one of the largest aqueducts in the world (completed in 2014) was not included in this map.



For implicit aqueducts, we considered that river water in a certain grid cell could be transferred to the neighboring cells if the following conditions were met. First, the origin and destination were in the same basin (i.e., no inter-basin water transfer). Second, the elevation of the destination had to be lower than the origin, which indicated that the gravity transfer of water was possible. To determine this, we used global elevation data from ETOPO1 Global Relief (Amante and Eakins, 2009). We upscaled from the original resolution of 1' × 1' into 0.5° × 0.5° by extracting the minimum elevation. Third, the river sequence of the destination had to be lower than that of the origin. The river sequence is a type of stream order assigned to every grid cell (see the numbers on boxes in Fig. 3). It takes the value one at the cell of headwater, and subsequently the value increases by one as the cell moves downstream. This condition was required to maintain the water balance of the river system, because the calculation of river routing in H08 is conducted in the order of the river sequence. The river routing calculation at the origin of aqueduct water transfer had to be conducted after the total volume of water transferred was fixed.

Because information regarding the capacity of aqueducts (i.e., the maximum rate of water transfer) was not available for most cases, we assumed that water could be transferred unless the river flow at the origin was depleted. Due to limitations in data availability, we assumed that both explicit and implicit aqueducts transfer water without any loss and delay. Hereafter, water withdrawal via aqueducts is expressed as *Waq* in all mathematical expressions.

### 2.1.4. Local reservoirs

To represent flow regulation by the major dams in the global river network, several algorithms have been devised (Hanasaki et al., 2006; Haddeland et al., 2006) and implemented in GHMs (Hanasaki et al., 2008b, Döll et al., 2009; Biemans et al., 2011). How best to represent minor reservoirs located in tributaries remains to be determined. We defined the term global reservoirs to be reservoirs located in the main channel of major rivers, which were explicitly delineated by the digital global river map used in the GHM, and defined local reservoirs as those located in the tributaries. A straightforward approach is to add the storage capacity of local reservoirs to that of global reservoirs (e.g., Wada et al., 2014), but this may overestimate the regulated flow capacity. Some studies have treated global and local reservoirs differently. The original H08 assumed that all reservoirs with less than 1 km$^3$ of storage capacity were local reservoirs (Hanasaki et al., 2010; they referred to them as "medium-size reservoirs"). Because the geographical information regarding local reservoirs was not available at that time, the authors spatially distributed the national total capacity of local reservoirs weighted by the population distribution. They assumed that local reservoirs were not regulating river flows, but acted as an ideal water storage location within grid cells. All the runoff generated in a grid cell runs into storage and the stored water can be used at any time. Wisser et al. (2010) introduced a similar algorithm for local reservoirs (they referred to them as "small reservoirs") into WBMplus (note that the abstraction from "small reservoirs" becomes unrealistically large, as much as 989 km$^3$yr$^{-1}$ in their formulations, i.e., one third of the total global water withdrawal).

We modified the original H08 algorithms for global and local reservoir operation (i.e., Hanasaki et al., 2006; 2010) as below. A schematic diagram of global and local reservoirs is shown in Fig. 1.





First, the GRanD global inventory of reservoirs (Lehner et al., 2011) was used to identify the location of global and local reservoirs. Global and local reservoirs were distinguished by their catchment area. GRanD includes the specifications of 6852 reservoirs with a storage capacity larger than 0.1 km$^3$, and all of them are georeferenced on the digital river network of HydroSHEDS (Lehner et al., 2008) at a spatial resolution of 15 arc-second. This enabled us to estimate the catchment area of

all the reservoirs, which has seldom been performed in previous global inventories of large dams. We set the threshold of 5000 km$^2$ (equivalent to the area of approximately two 0.5° × 0.5° grid cells) to separate global and local reservoirs. Note that the shape of the watershed and channel was not well reproduced in the digital river map of 0.5° × 0.5° for basins less than 5000 km$^2$. This resulted in 963 reservoirs with 4773 km$^3$ of total storage capacity being categorized as global reservoirs, and the remaining 5824 reservoirs (1300 km$^3$) being categorized as local reservoirs. In cases where multiple reservoirs were assigned

to one cell, their capacity was aggregated in each grid cell. The catchment area of a local reservoir was equal to that of the largest storage area within a grid cell, unless the area did not exceed the area of the grid cell.

Runoff generated within the catchment area of a local reservoir flowed into storage. When the storage exceeded its storage capacity, the excess water flowed into a river (Fig. 1). The storage in a local reservoir acted as an ideal tank, with water loss due to surface evaporation and other factors ignored. A local reservoir was accessible from the grid cells where it was located.

It was also accessible from the downstream grid cells connected by rivers and aqueducts. The water balance of local reservoirs is expressed as:

$$\frac{Slres}{dt} = Qtot \cdot Alres - Wlres - Qlres \tag{7}$$

where $Slres$ is the storage of local reservoirs [kg], $Qtot$ is the total runoff [kg m$^{-2}$ s$^{-1}$], $Alres$ is the catchment area of a local reservoir [m$^2$], $Wlres$ is the withdrawal-based abstraction from local reservoirs [kg s$^{-1}$], and $Qlres$ is the outflow from a local

reservoir that flows directly into the river channel of the cell [kg s$^{-1}$]. $Qlres$ is expressed as:

$$Qlres = max((Slres - Slres_{max})/\Delta t, 0) \tag{8}$$

where $Slres_{max}$ is the storage capacity of a local reservoir.

### 2.1.5.   Seawater desalination

Seawater desalination is a practical method to obtain freshwater in arid coastal regions. It currently accounts for approximately

0.1% of the total water withdrawal in the world, but production is rapidly increasing in arid regions (www.desaldata.com). Desalination was not incorporated in most of GHMs until recently. Oki et al. (2001) included the reported values of desalinated water production in total freshwater resources in a nation-based water scarcity assessment. Wada et al. (2011) spatially distributed the reported national volume of desalinated water produced along the coastline and assumed it was an available freshwater resource in the PCR-GLOBWB model. Recently, Hanasaki et al. (2016) developed a model to estimate the

geographical extent of areas where desalinated seawater is used and the volume of production. Because they produced a stand-alone model, incorporation of the scheme into a GHM is difficult.



We incorporated the seawater desalination scheme of Hanasaki et al. (2016) into H08. Their seawater desalination scheme consists of two parts. The first estimates the geographical extent of the area utilizing seawater desalination (AUSD), where seawater desalination is likely to be the dominant local water source. The second estimates the volume of water production. Hanasaki et al. (2016) found that the AUSD can be defined as all grid cells meeting all of three conditions, namely, the nations whose gross domestic product (GDP) exceeds 14,000 USD person$^{-1}$ yr$^{-1}$ in terms of purchasing power parity (PPP), the humidity index (precipitation over potential evapotranspiration) falls below 8%, and the cells are located within three consecutive $0.5° \times 0.5°$ grid cells (approximately 165 km along the equator) of seashore. By assuming seawater desalination is not used for irrigation, and all of the municipal and industrial water withdrawal in AUSD cells is abstracted by seawater desalination, which is supported by the available statistical records in Hanasaki et al. (2016), we could estimate the quantitative spatiotemporal distribution of withdrawal from seawater desalination. Hereafter, water withdrawal of seawater desalination is expressed as *Wdes* in mathematical expressions.

### 2.1.6. Return flow and delivery loss

Return flow and delivery (conveyance) loss are important processes in water abstraction. Döll and Siebert (2002) reported that of the 2549 km$^3$yr$^{-1}$ of global total water withdrawn for irrigation, 1092 km$^3$yr$^{-1}$ is used for consumption (evapotranspiration), which indicates that nearly 60% of water withdrawn for irrigation becomes return flow and delivery loss. We defined return flow as the flow of water that is withdrawn from sources, but is not consumed and is discharged into the original or some other water body. Delivery loss is the flow of water that is evaporated during delivery. LPJmL includes the return flow and delivery loss of irrigation (Rost et al., 2008). In their model, a certain fraction of abstracted water is delivered to cropland, which is determined by the regional irrigation efficiency. The authors then assumed that 50% of the undelivered water returns to the river channel, and the remaining 50% is lost due to evaporation.

To represent return flow and delivery loss for surface water abstraction, the algorithm of Rost et al. (2008) was adopted. Water abstraction is expressed as follows:

$$W = C + L + R \tag{9}$$
$$C = eW \tag{10}$$
$$L = l\ (1-e)W \tag{11}$$
$$R = (1-l)(1-e)W \tag{12}$$

where $W$ is withdrawal-based water abstraction taken from various sources [kg s$^{-1}$], $C$ is the consumptive water use, or the volume of water evaporated or transpired at the destination [kg s$^{-1}$]. $L$ is the delivery loss or the volume of water evaporated during delivery [kg s$^{-1}$], $R$ is the return flow (drainage) to the river stream [kg s$^{-1}$], $e$ is the ratio of consumption to withdrawal [-], and $l$ is the proportion lost during delivery [-]. $W$, $C$, $L$, $e$, and $l$ are all sector and water-source specific. Water lost through percolation was included in return flow. Return flow was drained into the subsequent downstream grid cell in the next time step (i.e. next day in a standard model set up).





The coefficient *e* for each sector was determined as follows. For surface water irrigation, the irrigation efficiency rate provided by Döll and Siebert (2002) was used. For groundwater irrigation, the rate was set at unity globally; we assumed that the irrigation wells are all located near to where the water was used. This does not necessarily mean that the abstracted groundwater was all consumed by evapotranspiration. As described in Appendix A, in H08, irrigation water is added to the soil moisture of

the irrigated portion of a grid cell. A substantial portion of it eventually turns into subsurface runoff and groundwater recharge, which is not used for evapotranspiration by plants. For industrial and municipal water, this proportion was set at 0.1 and 0.15 based on work by Shikilomanov (2000) for both surface and groundwater. To the best of our knowledge, there is no systematic global inventory of the amount lost during delivery (*l*). Following Rost et al. (2008), for surface water irrigation, we assumed the proportion lost was 0.5 globally. For groundwater irrigation, we assumed there was zero loss globally because we assumed

the distance of delivery to be negligible. Industrial and municipal water is drained underground; hence, we also assumed the loss to be zero globally, because water is seldom lost by evaporation, at least in the urbanized areas where the majority of industrial and municipal water is used.

### 2.1.7.  Surface water balance

Surface water abstraction was represented as follows. To fulfil the surface water requirement, water was first taken from local

river flow, which was regulated by a global reservoir. If the river was depleted and the surface water requirement was not fulfilled, water was taken from the river flow at the origin of an aqueduct. If the surface water requirement was still not met, water was taken from a local reservoir in the same cell or from an upstream location. If the grid cells in question were categorized as the AUSD (see Section 2.1.5), the surface water requirement for industrial and municipal water was taken from seawater desalination, and neither river flow nor local reservoirs were used. If the surface water requirement was still not met,

the remaining volume was classed as the surface water deficit or the volume of water that was unable to be abstracted from available surface water sources. Water was abstracted by sector in the order of municipality, industry, and irrigation.

The H08 model provides two modes of surface water deficit. Option 1 is to secure the fulfilment of the water requirement by assuming an imaginary unlimited surface water source and taking water from it as necessary. This is referred to as water abstraction from unspecified surface water (USW). Option 2 is to abandon the fulfilment of the water requirement and leave

the remaining volume as a deficit. Option 1 places more reliance on the estimated water requirement, which is largely statistically based or well validated in this study. It is difficult to explain where this volume of surface water comes from, which is a key shortcoming of this option. In contrast, option 2 places less reliance on the estimated water requirement. In this study, we used option 1. The estimated volume of abstraction from USW should be interpreted with care, and further consideration is given in Section 3.4.

The incorporation and coupling of all the schemes enabled H08 to express water abstraction as follows:

$$Wtot = Wgw + Wsrf \qquad (13)$$

where *Wtot* and *Wsrf* are total and surface water withdrawal, respectively [kg s$^{-1}$]. *Wsrf* is expressed as,

$$Wsrf = Wriv + Waq + Wlres + Wdes + Wusw \qquad (14)$$



where *Wriv* and *Wusw* are the withdrawal-based abstraction from a river and USW [kg s$^{-1}$], respectively. When Option 1 is taken, the following relationship is established:

$$Wsrf = (1 - f_{gw}) \times Qreq \qquad (15)$$

## 2.2. Data

### 2.2.1. Geographical data

Various geographical maps were used to set the boundary condition of the sub-models of H08, as shown in Table 1. We used the same setting as Hanasaki et al. (2013b) in this study. The geographical data used covered the whole globe, except Antarctica, at a spatial resolution of $0.5° \times 0.5°$ for the year 2000.

The geographical data directly relevant to the key results are described here. The irrigation water requirement was substantially influenced by the irrigated area (Siebert et al., 2010), crop type (Monfreda et al., 2010), crop intensity, and irrigation efficiency (Döll and Siebert, 2002). The industrial and municipal water requirement was determined by national estimates for the year 2000 provided by AQUASTAT (www.fao.org/nr/aquastat/). The national estimates were spatially interpolated and weighted by the population distribution of the Center for International Earth Science Information Network (CIESIN) and Columbia University and Centro Internacional de Agricultura Tropical (CIAT) (2005).

### 2.2.2. Meteorological data

To run H08, a global meteorological dataset is required. We used the WATCH forcing data methodology applied to European Centre for Medium-Range Weather Forecasts (ECMWF) re-analysis (ERA)-Interim data (hereafter WFDEI, Weedon et al., 2014) in this study. WFDEI contain eight meteorological variables, namely, air temperature, specific humidity, wind speed, surface air pressure, longwave and shortwave downward radiation, rainfall, and snowfall. WFDEI covers the whole globe at a spatial resolution of $0.5° \times 0.5°$, and the period of 1979-2013 at a daily interval.

## 2.3. Simulations

Simulations were conducted for 35 years from 1979 to 2013 using WFDEI. The simulations for the last 30 years were used for analyses (1984-2013), with the first five years used only as a spin-up. The geographical data were fixed at the year 2000 due to the limited availability of temporal variations. The calculation interval was a day. This means that all the water flows accompanying both natural and human processes were calculated and exchanged among components at a daily interval, strictly maintaining the water balance.

Two simulations were conducted in this study. The first was a naturalized simulation, which disabled all of the sub-models of human activity (NAT). For this simulation, only the land surface hydrology model enhanced by the groundwater recharge scheme and the river routing sub-models were used. The second was a simulation that enabled all sub-models of human activities to be enhanced by the abovementioned six schemes (ALL). For this simulation, the daily grid-based water



requirement was fulfilled by any one of the six explicit water sources (i.e., renewable groundwater, nonrenewable groundwater, river, aqueduct water transfer, local reservoirs, seawater desalination) or USW.

## 3. Results and discussion

The simulation results were investigated from four viewpoints. First, to validate the newly added schemes, we compared the
simulation outputs with reliable earlier estimates. Second, to assess the overall performance of the hydrological simulation, we compared the simulated river discharge and terrestrial water storage (TWS) with observed flow records and a satellite product from the GRACE mission, respectively, for selected major global basins. Third, to achieve the primary objective of this study, we investigated the source of water withdrawal at the global and regional scale. Finally, the uncertainty in the model was given extensive consideration.

**3.1. Results with the new hydrological components added**

Table 2 shows a comparison of global estimates of mean annual groundwater recharge, total groundwater withdrawal, nonrenewable groundwater withdrawal, return flow, delivery loss, and abstraction from local reservoirs with those in earlier works. Figure 5 shows the global distribution of these terms. Figure 6 shows a comparison of national estimates with those reported in a recent study.

**3.1.1. Groundwater recharge**

The mean annual groundwater recharge was estimated to be 13466 km$^3$ yr$^{-1}$. This estimate was within the range of those reported in earlier simulation-based studies (12666-15200 km$^3$ yr$^{-1}$; Table 2) and agreed well with the statistics-based report by IGRAC (2004). Although the method adopted to estimate groundwater recharge in this study was identical to that of Döll and Fiedler (2008), the results differed mainly because of the difference in runoff estimates (i.e., $Q_{tot}$ in Eq. 1).
The spatial distribution of groundwater recharge (Fig. 5 a) was fundamentally similar to the total runoff (Fig. S1 a), but it also reflected the five groundwater factors shown in Eq. 1. In terms of the broad spatial pattern, it agreed well with earlier works, for example, Fig. 5 of Döll and Fiedler (2008) and Fig. 1 of Wada et al. (2010). The key characteristics of this study were the high rate of recharge in northern Europe, western Siberia, and eastern Canada, and low rate of recharge in southern China compared to the results of two earlier studies.
Figure 6 (a) is a scatter plot comparing the mean annual national groundwater recharge estimated by our model to that reported by IGRAC (2004). Among the countries where the annual recharge exceeds 500 km$^3$ yr$^{-1}$, the errors of Brazil and Colombia were less than ±20%, and those for USA, Russia, and China were between -50 and +100%. Even greater differences between the two estimates were apparent for some countries (e.g. Indonesia, Canada, Australia, Angola, New Zealand), which was not surprising given the fundamental difficulty in estimating groundwater recharge precisely with any method (e.g., Healy, 2010).





The spread of results was much smaller when the estimates were compared with Döll and Fiedler (2008), who adopted the same methodology (not shown).

### 3.1.2.    Groundwater withdrawal

The mean annual global total groundwater withdrawal was estimated to be 789 km$^3$ yr$^{-1}$ in this study. This estimate was within

the range reported in earlier studies (710-952 km$^3$ yr$^{-1}$; Table 3). Similar to groundwater recharge, the spatial distribution of groundwater withdrawal also agreed well with the distribution reported in the earlier studies by Wada et al. (2014) and Döll et al. (2012). Groundwater is most intensively used in central USA, northwestern India, and northern China, which was clearly reproduced in our results (Fig. 5 b). Figure 6 (b) is a scatter graph comparing the mean annual national groundwater withdrawal estimated by our model and that reported by IGRAC (2004). They agree well, particularly for major countries such as India,

China, USA, and Pakistan.

Groundwater withdrawal is conducted to satisfy the demand of three sectors, namely irrigation, industry, and the municipality. As shown in Table 3, the estimations for industrial and municipal use agree well with those of IGRAC (2004). This is mainly because the numbers are largely statistically dependent. The national total water withdrawal for these sectors was taken from AQUASTAT, and the groundwater fractional contribution was derived from IGRAC (2004). In contrast, agricultural water,

which is the dominant water use for most of the major groundwater-using countries, is largely model dependent, because the crop calendar and irrigation application were both simulated (see Appendix A). The agreement between earlier estimates, particularly at the national level, implies the validity of our model.

Abstraction from nonrenewable groundwater reservoirs results in groundwater depletion, which was estimated to be 182 km$^3$ yr$^{-1}$ in this study (Table 2). This agrees well with statistics-based studies (145-200 km$^3$ yr$^{-1}$; Postel, 1999; Konikow et al., 2011)

and was within the range of the results from the latest simulation based studies (113-330 km$^3$ yr$^{-1}$; Wada et al., 2014; Döll et al., 2014; Pokhrel et al., 2015). The considerable range of results among the earlier studies reflects the limitations in reliable in-situ data (e.g., Wada, 2016). The range of simulation-based estimates was particularly large because it was basically estimated by the difference between the rate of groundwater recharge and the groundwater requirement, and both parameters contain substantial uncertainties. The spatial distribution of groundwater depletion was concentrated in specific regions of the

world, including the High Plains Aquifer in the USA, the North China Plain, western India and a part of eastern Pakistan, and the central Arabian Peninsula (Fig. 5 c). These areas clearly overlapped with the areas where groundwater irrigation is required in arid to semi-arid regions (Fig. 2a).

Figure 6 (c) shows a comparison of the national volume of groundwater depletion estimated by this study and Döll et al. (2014). The estimated volume of nonrenewable groundwater usage for major countries in this study tended to be larger than in Döll et

al. (2014), except for the USA. In particular, the estimations for India and China were more than double the estimates of Döll et al. (2014).



### 3.1.3.  Aqueduct water transfer

Aqueduct water transfer was estimated to be 199 km$^3$ yr$^{-1}$ globally. This is hard to validate because, to the best of our knowledge, a similar number has not been reported elsewhere. It should also be noted that the difference between water abstraction from a "river" and an "aqueduct" is only the distance of water transfer (taking water from within a grid cell or its neighbors). Even

if a reliable global dataset of aqueduct water transfer were available, it would be difficult to re-compile the data according to the distance of water transfer (approximately 55 km in this study), to distinguish between a "river" and an "aqueduct". The distribution of the volume of aqueduct water transfer is shown in Fig. 7 for four regions where water is heavily managed. It clearly indicates the major rivers in each region; in particular, the Indus and the Huang He Rivers supplied water to surrounding cells through implicit aqueducts (compare with Fig. 4). In some grid cells in southern coastal California and the

Nile Delta, water was transferred via explicit aqueducts depicted in Fig. 4.

### 3.1.4.  Local reservoirs

Abstraction from local reservoirs was estimated at 106 km$^3$ yr$^{-1}$ globally (Table 2). There are few other available studies that could confirm this number. Biemans et al. (2011) reported that reservoirs (both global and local) contributed an additional 460 km$^3$ yr$^{-1}$ of irrigation water supply globally at the end of 20$^{th}$ century. Taking the total capacity of local reservoirs in this study

(1300 km$^3$, which corresponds to 21% of the total capacity of reservoirs) into consideration, our estimate agrees fairly well with Biemans et al. (2011).

### 3.1.5.  Seawater desalination

Seawater desalination was estimated to be 1.8 km$^3$ yr$^{-1}$. This is less than the estimate reported by Hanasaki et al. (2016) because of the difference in base year (i.e., 2000 for this study and 2005 for Hanasaki et al.). The estimate of AQUASTAT was

substantially larger (4.6 km$^3$ yr$^{-1}$) but it did include saline surface water and groundwater as a source. The distribution of the usage of seawater desalination is shown in Figure 8. As reported in Hanasaki et al. (2016), 85% of the world's seawater desalination use is concentrated in nine countries, namely, United Arab Emirates, Saudi Arabia, Kuwait, Spain, Qatar, Libya, Bahrain, Israel, Oman. The distribution overlaps with the coastal area of these countries, except for Spain. Major seawater desalination plants in Spain are located on the southeast coast where the climate is more humid than the other eight countries.

Due to limitations of the seawater desalination scheme, seawater desalination in regions with relatively humid climate is not successfully reproduced. See Hanasaki et al. (2016) for further discussion.

### 3.1.6.  Return flow and delivery loss

Global return flow and delivery loss was estimated to be 1546 and 590 km$^3$ yr$^{-1}$, respectively (Table 2). These numbers are similar to the estimates by Jägermeyr et al. (2015), who adopted the same assumptions for return flow and delivery loss (Rost

et al., 2008). Both return flow and delivery loss were spatially concentrated in Asia, where surface irrigation is predominant



(Figs. 5 d-e and 2 a). This primarily reflects the practice of paddy irrigation, which requires large amounts of irrigation water to flood the paddy field, with an accompanying low irrigation efficiency.

## 3.2. Validation at selected basins

To assess the influence of the six schemes in global hydrological simulations, we validated the simulated river discharge and

TWS against in-situ and satellite observations.

We investigated the results for twelve of the world's major basins. First, we selected the ten largest basins in the world, namely the Amazon, Congo, Mississippi, Parana, Nile, Yenisei, Ob, Lena, Chang Jiang, and Amur rivers. We excluded the Nile River from the investigation, because its discharge was considerably overestimated. It is frequently reported that GHMs substantially overestimate the river discharge of the Nile River (e.g., Haddeland et al., 2011; Hattermann et al., 2017). This poor performance

for the Nile River by H08 was attributed not only to the model's formulation but also to the reliability of meteorological data in the basin, which has been commonly seen in other GHMs. Among the ten basins, the Mississippi, Parana, and Chang Jiang rivers are the most heavily influenced by human activities. We then added the Ganges, Colorado, and Huang He rivers. These are large river basins where considerable water management occurs, and for which river discharge observations were available for more than five years. We considered these six basins to be heavily human-affected basins. The remaining six rivers were

considered to be less heavily human-affected basins. We focus on the results for the heavily human-affected basins in this subsection. The results for the less heavily human-affected basins are shown in the Supplemental Material.

### 3.2.1.    River discharge of heavily human-affected basins

We validated the simulated river discharge at the major gauging station of six heavily human-affected basins using the observation records provided by the Global Runoff Data Centre (www.bafg.de/GRDC/EN/Home/homepage_node.html). The

river discharge observations at Hardinge Bridge for the Ganges River were obtained from Masood et al. (2015).

In all six basins, there were notable differences between the ALL and NAT simulations. The differences were primarily due to two effects. One was the effect of reservoir operation, leading to a diminishing of the seasonal variation in river flow. The peak flow was lowered, and the low flow was increased, by storing water during wet periods and releasing it in dry periods. The other was the effect of water abstraction, leading to a decrease in river flow. The effect was dominant during periods of

low flow, which was typically when the irrigation water requirement was concentrated. In four of the six basins, the ALL simulation outperformed NAT in terms of the Nash-Sutcliffe efficiency (NSE; Nash and Sutcliffe, 1970) and the bias (Table 4). Here the values of NSE and bias were used to compare NAT and ALL simulations. Since the hydrological parameters of H08 were not tuned at individual basins, the results of NSE and bias in some basins were not as high as those of calibrated models (see Hattermann et al. 2017 for comprehensive discussion).

For the Mississippi River, both the ALL and NAT simulations reproduced the historical observations well  (Fig. 9 a). The ALL simulation performed better than NAT due to the inclusion of the reservoir operation effect (i.e., a decrease in the peak flow and an increase in the low flow). For the Parana River, although the period of validation was quite short due to data



limitations (1979-1983) there was a tendency to overestimate discharge. The ALL simulation performed substantially better than NAT, and reproduced the observations fairly well. Again, the effect of reservoir operation played an important role, with the basin heavily regulated by a number of global reservoirs. For the Chang Jiang River, the ALL simulation performed slightly worse than NAT. The simulation had a tendency to underestimate discharge in this basin. In this case, the effect of water
abstraction exacerbated the negative bias and NSE. For the Ganges River, the ALL simulation performed better than NAT. This good performance was due to the effect of reservoir operation, which reduced the peak flow and the effect of water abstraction in reducing the low flow for irrigation. For the Huang He River, the ALL simulation performed slightly worse than NAT. Again there was a tendency to underestimate discharge. The large water withdrawal, which was as much as 34% of the mean annual river discharge, exacerbated the underestimation. Finally, for the Colorado River, the ALL simulation performed
substantially better than NAT. The simulated river discharge displayed a step-wise temporal variation every 12 months, which was due to the reservoir operation algorithm implemented in H08 (see discussion in the Supplemental Text). The gauging station was located just below the Glen Canyon Dam, and river discharge reflected the operation of the dam (see Fig. S5 b).

### 3.2.2. TWS in heavily human-affected basins

TWS is the water stored on and beneath the land surface, which consists of seven components in H08. Six of these are the
changes in water storage of soil moisture, snow, renewable groundwater, river water, water storage in global reservoirs, and local reservoirs. The seventh component is simulated groundwater depletion or the accumulation of abstraction from nonrenewable groundwater reservoirs over time. To validate TWS, we use the TWS anomaly (TWSA), which is the monthly mean difference from the long-term mean annual TWS. TWSA reflects both the seasonal and inter-annual variations in TWS. To validate the simulated TWSA, we used monthly gridded TWS data derived from GRACE. The GRACE products we used
were the spherical harmonic solutions (Level-3, Release-5) of equivalent water height thickness processed by the Center for Space Research (CSR) at the University of Texas, Austin (Landerer and Swenson, 2012).

Figure 10 includes the TWSA and TWS components of heavily human-affected basins. The results for less heavily human-affected basins are shown in Fig. S4. Overall, the model reproduced the distinct monthly peaks seen in the TWSA of the GRACE products fairly well, together with the amplitude in the seasonal variations in TWSA, which differed substantially
among the basins. For example, the Ganges River had an amplitude of ± 200 mm (Fig. 10d), while the amplitude for the Huang He River was only ± 50 mm (Fig. 10 e).

In four out of six heavily human-affected basins, the ALL simulation outperformed NAT in terms of the NSE, correlation coefficient, and the trend in TWSA (Table 4). The differences between the ALL and NAT simulations were attributed to the anthropogenic terrestrial water component of global and local reservoirs and groundwater depletion. The exact reasons for the
good performance in the ALL simulation compared to NAT varied among basins. For the Parana River, incorporation of the reservoirs contributed to the good performance. The basin is heavily regulated by global reservoirs, and this water storage is one of the largest terrestrial water components of this basin (Fig. 10 b). Groundwater depletion is negligible because there are few irrigated areas equipped with groundwater abstraction in this basin. For the Ganges River, substantial groundwater



depletion was simulated (approximately 20 mm yr$^{-1}$; Fig. 10 d). Although this was larger than the rate of depletion reported by GRACE (10 mm yr$^{-1}$), the inclusion of groundwater depletion contributed to the downward trend in the TWSA of the basin, together with fairly well-reproduced inter and intra annual variations. For the Huang He River, both water storage in the global reservoirs and groundwater depletion were the dominant components (Fig. 10 e). Similar to the Ganges River, the inclusion of

groundwater depletion enabled the downward trend of TWSA to be reproduced. The dominant terrestrial water component in the Colorado River was the water stored in the global reservoirs (Fig. 10f). Groundwater depletion was estimated to be marginal in this basin. It is interesting to observe that the decreasing trend of TWSA was largely attributed to global reservoirs from 2005. In other words, while the downward trend seen in the Ganges and the Huang He Rivers could be explained by groundwater depletion, the trend in the Colorado River was explained by the trend toward water storage in global reservoirs.

The ALL simulation underperformed compared to NAT in two basins. The Mississippi River was moderately regulated by global and local reservoirs (Fig. 10 a). Groundwater depletion was estimated to be approximately 7 mm yr$^{-1}$ and was concentrated in the High Plains Aquifer as mentioned earlier. The rate of depletion was substantially larger than that reported by GRACE (0.50 mm yr$^{-1}$), and, consequently, it exacerbated the correlation coefficient and NSE. For the Chang Jiang River, the performance was slightly worsened by the incorporation of human activity, but the difference was trivial (Table 4). The

basin was moderately regulated by global and local reservoirs and groundwater depletion was negligible. The predominant TWS component was river water storage (Fig. 10 c). Although the basin is densely populated, human inventions were relatively minor compared with the natural hydrological components.

Although we confirmed that simulated global and national estimates in groundwater depletion agree fairly well with reported values, the comparison with GRACE implies a general tendency for overestimation. This point is revisited in Section 3.4.

**3.3. Water abstraction by sources**

Table 5 shows the mean annual global total volume of water abstraction by sources and sectors. Figure 1 also includes selected variables for global water cycle and abstraction by humans. Surface water is the dominant water source globally, accounting for 2839 km$^3$ yr$^{-1}$ or 78% of the total water withdrawal. Groundwater accounts for the remaining 789 km$^3$ yr$^{-1}$ or 22%. These numbers agreed well with the earlier estimates by WaterGAP (Döll et al., 2012) and PCR-GLOBWB (Wada et al., 2014).

Total, surface, and groundwater water withdrawal were very close to the values obtained from AQUASTAT and IGRAC (2004).

The enhanced H08 enabled us to break down these estimates further in terms of water sources and sectors. For irrigation water, more than half of the surface water consumed comes from rivers. Aqueducts and local reservoirs also make important contributions (9 and 5%, respectively). No seawater desalination was used because it was assumed it was not used for irrigation.

USW (see Section 2.1.7) was very large in the irrigation sector. It accounted for 33% of the annual total agricultural water withdrawal. This large volume of USW is discussed in Section 3.4.1. As for groundwater, 31% was obtained from nonrenewable sources. For industrial and municipal water use, river water withdrawal accounts for most of the surface water supply (85 and 90%, respectively). The majority of surface water comes from rivers in our simulation. As shown earlier,



seawater desalination provided 0.4 and 1.4 km$^3$ yr$^{-1}$ of industrial and municipal water, respectively. Although the numbers are small, this can sustain arid regions where the availability of alternative water sources is limited. The fractional contributions of USW and nonrenewable groundwater withdrawal for these sectors were substantially smaller than those for irrigation (10 and 6% for industrial water and 7 and 5% for municipal water, respectively). This was partly due to the order of water

abstraction. As described in Section 2.1.7, water was abstracted from sources in the order of seawater desalination (only for industrial and municipal use in limited areas), rivers, aqueducts, and local reservoirs, and for sectors in the order of municipalities, industry, and irrigation. Municipalities have more opportunities to obtain river water than other sectors.

For reference, the simulation outputs of the original H08 are also shown in Table 5. Total water consumption (not withdrawal, see Appendix A) for all sectors was estimated to be 1466 km$^3$ yr$^{-1}$. River water and medium-sized reservoirs (i.e., local

reservoirs in this study) supplied 541 and 494 km$^3$ yr$^{-1}$, respectively, and the remaining 432 km$^3$ yr$^{-1}$ was supplied from nonlocal and nonrenewable blue water (NNBW; i.e., the sum of the abstraction from nonrenewable groundwater and the surface water deficit in this study). Although the estimated total consumption compared well with earlier simulation-based estimations, further validation was hampered for two reasons. One was the availability of validation data or consumption-based statistics regarding water use. The other was that the highly idealized and conceptual water-source components in the original H08 were

hard to interpret. For example, because the source of water was not separated into surface and groundwater, Hanasaki et al. (2010) explained that renewable and nonrenewable groundwater was "implicitly" included in river water and NNBW, respectively. These problems were fully addressed by the incorporation of the six schemes in H08.

Figure 11 shows the estimated fractional contributions of water sources for 21 regions defined by Giorgi and Francisco (2000). See Table S2 for the list of regions. For all regions, river water was the source that made the largest contribution. Aqueduct

water transfer played an important role in Central Asia (CAS) and the Mediterranean (MED), which was due to irrigation surrounding major rivers such as the Nile, Amu Darya, and Syr Darya. Note that CAS included a part of the upper Indus river basin where a dense aqueduct network exists (see Fig. 4 d and Fig. 7 d). The fractional contribution of seawater desalination was only notable in the Sahara (SAH), which includes the Arabian Peninsula where most of the world's seawater desalination plants were concentrated in 2000. The fractional contribution of groundwater (the red arc) was particularly large in Central

North America (CAN), Central America (CAM), SAH, and South Asia (SAS). The fractional contribution of nonrenewable groundwater, seawater desalination, and USW (the black arc) was particularly large in CNA, SAH, SAS, East Africa (EAF), CAS, and Tibet (TIB). All of these are arid or semi-arid regions. Note that TIB included a major part of the Xinjiang Uyghur Autonomous Region in China and part of northwestern India, with both having an arid and semi-arid climate and a vast irrigation area. The black arc is marginal only for the regions in northern latitudes, such as Alaska (ALA), Greenland (GRL),

Northern Europe (NEU), and North Asia (NAS). The results indicate that further investigation is needed on the consistency between the simulated water availability and use in mid to low latitudes.





### 3.4. Uncertainties

Although H08 has been substantially improved by incorporating the six schemes, uncertainties still remain. In this subsection, we summarize the key uncertainties and challenges, with a particular focus on the terms USW and nonrenewable groundwater.

**3.4.1.    Unspecified surface water (USW)**

The key precondition of H08 (and to the best of our knowledge all of the GHMs with human water abstraction) is that the water requirement is first determined and, subsequently, water withdrawal from specific sources is estimated taking spatiotemporal water availability into account. Hence, USW can be regarded as the inconsistency between the water availability and water requirement. USW was simulated to be 747 $km^3yr^{-1}$ by introducing USW or taking option 1 (Section

2.1.7). As shown in Table 5, the simulated total water withdrawal (3628 $km^3yr^{-1}$), which is identical to the water requirement when option 1 is taken for USW, was close to the reported global water withdrawal (3550 $km^3yr^{-1}$, AQUASTAT) and the fractional contributions of surface water and groundwater were also close to the ratio reported by IGRAC (2004). However, in the H08 simulation the water requirement is not fulfilled by accessible surface water.

Figure 12 (a) shows the global distribution of USW. USW was extensively distributed in the Indian subcontinent, China, and

western North America, particularly concentrated in northern India, Pakistan, and coastal northern China. The majority of USW was attributed to the shortage in surface irrigation water (Table 5).

To understand more fully the mechanism of why USW was needed, we undertook a further investigation of the water components at one specific point. Figure 13 shows the mean monthly water balance in a grid cell near Bangkok, Thailand, (N13.25° E100.25°). The location was characterized by distinct dry and wet seasons from May to October and from November

to April, respectively (Fig. 13 a). The seasonal variation in river flow reflected that of precipitation. The temporal variation in the water requirement clearly displayed the opposite pattern, with a concentration in the dry season (Fig. 13 c). Although the global reservoirs in upstream locations released water intensively during this period (see Fig. S5 d), river water was not sufficient to supply the water requirement. Water stored in the local reservoirs in upstream locations was also not sufficient; hence, the water supply was depleted in the dry season (Fig. 13 b). Eventually, the water requirement assigned to surface water

fell short and USW was recorded for this grid cell (Fig. 13 c).

Figure 12 (b) shows the area supplied by global and local reservoirs. It shows the total storage capacity of all global and local reservoirs in upstream locations divided by the number of seconds in a year, or the rate of flow if the reservoirs released all of the stored water constantly during a year. The area supplied is extensive; however, compared with the area of USW (Fig. 12a), the water stored in the reservoirs was not sufficient to fully meet the USW. For example, the USW was concentrated in

northwestern India and northeastern China, while only a few local reservoirs were located in these regions.





### 3.4.2. Nonrenewable groundwater

The estimation of the volume of nonrenewable groundwater contained considerable uncertainties. Although the estimated volume of global total nonrenewable groundwater agreed well with earlier estimates (113-330 km$^3$yr$^{-1}$; Table 5), the simulated trends of TWSA for selected basins, as shown in Fig. 10, tended to be overestimated. This contradiction was largely attributed to the uncertainty in both the quantitative and geographic estimation of nonrenewable groundwater usage. As shown in Fig. 6c, the national estimates still differed substantially from the earlier simulation-based estimates. These results imply that nonrenewable groundwater might be overestimated to some extent.

### 3.4.3. Potential sources of uncertainty

There were four major sources of uncertainty in this study, which, potentially, caused the paradoxes of USW and nonrenewable groundwater. The first was the limitation in the performance of physical hydrological sub-models. In particular, the rate of river flow in the dry season and groundwater recharge influenced the simulation of water availability. Although the key hydrological processes were represented, the land surface hydrology and river routing sub-models of H08 are relatively simple (Appendix A). Moreover, the hydrological parameters were not tuned to individual basins, which yielded a generally lower reproducibility of historical river flow observations (e.g., Hattermann et al., 2017). Also, the global climate data we used contained uncertainties. There are still considerable discrepancies among the latest global meteorological data sets (e.g., Müller-Schmied et al., 2014), which implies that there are regions where the input data are likely to be not well constrained by observations.

The second source of uncertainty was that H08 still omits some important water sources. They include, small ponds and reservoirs, and melt water from glaciers. In this study, we accounted for 6852 major reservoirs, totaling 6197 km$^3$ of storage capacity, which were registered in the GRanD database as global and local reservoirs. Lehner et al. (2011) estimated that the total number and water storage of reservoirs in the world is 16.7 million and 8070 km$^3$, respectively, which implies approximately 16 million and 2000 km$^3$ minor ponds and reservoirs are still not accounted for in this study. No database of such ponds and reservoirs has been developed yet, but if they were included, the temporal gap between water requirement and availability would be further diminished. Glacier melt water, which was not considered in this study, might increase surface water availability in some regions of the world, typically central Asia. Hirabayashi et al. (2010) estimated the annual global total melt water from glaciers to be 19.8 km$^3$ yr$^{-1}$ (1990-2003). In addition, improvements are also needed for the aqueduct database. Although aqueducts were considered in the model, the number and coverage of explicit aqueducts was less than the actual situation.

The third source of uncertainty was that the models and algorithms for the water requirement. Because it is by far the largest water user of the three sectors considered in the study, the estimation of irrigation water is crucially important. Probably the largest assumption in H08 is that the irrigation water requirement is fully met (i.e., soil moisture is kept at a certain level; see Appendix A) during the cropping period. This may overestimate the irrigation water requirement in arid and semi-arid climates



where deficit irrigation is practiced (Döll et al., 2014). We fixed the irrigated area throughout the simulation period but, in reality, it may vary according to local water availability. Further incorporation of local agricultural practices into the model is crucial to improve its performance. This includes crop type, cultivars, planting, and harvesting date, timing and intensity of irrigation, permitted water use, and irrigation equipment, all of which affect the temporal variation and volume of the irrigation water requirement.

The fourth source of uncertainty regards spatiotemporal resolution, which should be further investigated. It was a fundamental presumption of this study that the minimum temporal and spatial unit of hydrology and water use was one day and one $0.5° \times 0.5°$ grid cell. A further increase in spatiotemporal resolution would enhance the use of return flow, which would increase the gross water availability.

## 4. Conclusions

Six schemes were added to the H08 model to represent human water abstraction more accurately and ensure the water balance in each grid cell at a daily interval. Our model indicated that in the year 2000, of a total freshwater abstraction of 3628 km$^3$yr$^{-1}$, 2839 km$^3$yr$^{-1}$ was from surface water and 789 km$^3$yr$^{-1}$ was from groundwater. Streamflow, aqueducts, local reservoirs, and seawater desalination accounted for 1786, 199, 106, and 1.8 km$^3$yr$^{-1}$ of the surface water, respectively, while 747 km$^3$yr$^{-1}$ of surface water requirement was unmet. Renewable and nonrenewable groundwater accounted for 607 and 182 km$^3$yr$^{-1}$, respectively. These estimations agreed well with earlier studies that were based on statistics and simulations. Furthermore, the incorporation of these six schemes improved the river discharge and TWSA simulations in heavily human-affected basins.

Every water source differed in its renewability, economic costs for development, and environmental consequences for usage. To cope with the increasing water requirement and changing climate in the 21$^{st}$ century, efficient water management is crucially important all over the world. The enhanced H08 model can incorporate the aspects of sustainability, economy, and environment into forthcoming global water resource assessments.

Among the six schemes, local reservoirs, aqueduct water transfer, and seawater desalination were incorporated into GHMs for the first time, to the best of our knowledge. The key concept of a local reservoir is that of isolating the storage of reservoirs in tributaries from global river systems. This is a practical way to express the role of small reservoirs in the limited spatial resolution of a GHM. The aqueduct water transfer scheme enables water from major rivers to be distributed to surrounding grid cells. Because a comprehensive digital global inventory of aqueducts is not currently available, the scheme includes an algorithm to infer the potential routes of aqueducts. This scheme reduces the gap in water availability in the grid cells between periods when the major rivers are flowing and when they are not. This function is potentially useful in high-resolution water resource modeling (e.g., Wada et al., 2016), although inter-cell water transfer is a key technical issue that needs to be resolved. The seawater desalination scheme enables the intensive water-use taking place in some of the arid coastal regions (e.g., the Gulf countries) to be explained. Although the contribution of seawater desalination is currently limited in terms of quantity and spatial extent, it could expand considerably due to socio-economic conditions and climate change (Hanasaki et al., 2016).





**Acknowledgment**

This work was mainly supported by the Japan Society for the Promotion of Science (JSPS) KAKENHI Grant Number 16H06291. It was partially supported by JSPS KAKENHI Grant Number 25820230, the Environment Research and Technology Development Fund (S-14) of the Ministry of the Environment, Japan, and the Core Research for Evolutionary

Science and Technology (CREST), Japan Science and Technology Agency. The present work was partially developed within the framework of the Panta Rhei Research Initiative of the International Association of Hydrological Sciences (IAHS) by the Water Scarcity Assessment: Methodology and Application working group. The authors thank Yoshihide Wada for providing technical details regarding PCR-GLOBWB and Hodaka Kamoshida for technical assistance with handling GIS data. Map colors were based on www.ColorBrewer.org, by Cynthia A. Brewer, Penn State. Observed runoff data were kindly provided

by the Global Runoff Data Centre, 56068 Koblenz, Germany.

**Appendix A**

The H08 model is a grid-cell based GHM. Although the name H08 is used to refer to "a model described in Hanasaki et al. (2008a,b)", we also included additional sub-models described in Hanasaki et al. (2010 and 2013a,b) in H08. The H08 model has been used in several advanced global water-resource assessments to assess water scarcity globally, with consideration

given to the seasonality of water availability and use, in historical periods (Hanasaki et al. 2008b) and future periods under comprehensive global change scenarios (Hanasaki et al. 2013a,b). It has also been used to estimate the global flow of so-called 'virtual water', providing details of the sources of water and their temporal evolution (Hanasaki et al., 2010; Hanasaki, 2016; Dalin et al., 2012). The model has participated in major international model inter-comparison projects, such as EU Water and Global Change (EU-WATCH; Haddeland et al., 2011) and Inter-Sectoral Impact Model Intercomparison Project (ISIMIP;

Schewe et al., 2014; Haddeland et al., 2014), to identify its strengths and weaknesses among the other influential models that are currently available. Recently, H08 has been applied in regional domains, with calibrated hydrological parameters (Hanasaki et al., 2014; Mateo et al., 2014; Masood et al., 2015).

The original H08 model consists of six sub-models, namely land surface, river routing, reservoir operation, crop growth, environmental flow, and water abstraction. The formulations of these sub-models are described in detail in Hanasaki et al.

(2008a,b, 2010). In the standard simulation settings, H08 covers the whole globe at a resolution of 1° × 1° or 0.5° × 0.5° to assess the geographical heterogeneity of hydrology and water use. Typically, the simulation period is several decades and the calculation interval is one day. The six sub-models exchange water fluxes and update the water storage at each calculation interval, with a complete water balance (the error is less than 0.01% of the total input precipitation). These characteristics have enabled explicit simulations of the major interactions between the natural water cycle and major human activities around the

globe. The source code and H08 manuals are open to the public, and are available at http://h08.nies.go.jp.

The land surface sub-model is a bucket model (Manabe et al., 1969; Robock et al., 1995), with a simple drainage scheme that generates subsurface runoff (Gerten et al., 2004). It solves the energy balance of the land surface and the water balance of the



top 1 m of soil. It estimates surface runoff, which is generated when the soil is saturated, and subsurface runoff, which is generated regularly as a function of soil moisture. A land grid cell is subdivided into four sub-cells to represent the different land uses, namely, double-crop irrigated, single-crop irrigated, rainfed cropland, and non-cropland. The fractional contributions of each sub-cell are determined by the specific geographical maps listed in Table 1. The river routing sub-model

is a simple scheme to transport total runoff (sum of surface and subsurface runoff) through the digital grid-based river network (Oki et al., 1999). When a river section includes large reservoirs (LRs; corresponding to global reservoirs in this study), the river flow is regulated following the specific reservoir operation rule generated by the reservoir operation sub-model. LRs are defined as reservoirs with more than 1 km$^3$ of total storage capacity (Hanasaki et al., 2006). LRs, which total 507 in number and 4411 km$^3$ in storage capacity, are explicitly georeferenced in the digital river network of H08. The crop growth sub-model

plays two roles. One is to estimate the planting date of crops globally. This is done by the use of a stand-alone crop growth model prior to a H08 hydrological simulation, using the planting and harvesting date in a year that yields maximum crop production. The other role is to simulate crop growth during a H08 hydrological simulation, which provides the daily status of cultivation for use in the water abstraction sub-model. The environmental flow sub-model estimates the river flow that should be kept in the river channel for the aquatic ecosystem. The water abstraction sub-model, which was the main concern of this

study, is described in detail in the next paragraph.

Figure A1 shows a schematic diagram of the water abstraction sub-model of the original H08 model. It deals with water consumption in three sectors, namely, irrigation, industry, and municipality. Here water consumption is defined as water evaporated during the utilization of water. The H08 model considers water that is evaporated without any loss in delivery. Daily water requirements for irrigation are defined as the volume of water needed to maintain a certain level of soil moisture

during a cropping period. The volume of irrigation water is added to the sub-cell to represent the soil moisture of irrigated cropland. Industrial and municipal water requirements are prescribed (i.e., not dynamically simulated in H08). To meet the water requirement at each grid-cell at a daily interval, the water abstraction sub-model takes water from three sources: river flow with regulation by large reservoirs (hereafter RIV), storage in medium-size reservoirs (MSR), and nonlocal and nonrenewable blue water (NNBW). The water abstraction sub-model first takes water from RIV until it satisfies the water

requirement, or river flow reaches the environmental flow. When the water requirement is not fully met, water stored in MSRs is abstracted. MSRs are idealized representations of reservoirs less than 1 km$^3$, which total approximately 25000 in number and 2783 km$^3$ in storage capacity. Because individual MSRs are seldom located on the main stem of major rivers in reality, they are modeled differently from LRs. MSRs are aggregated by each grid cell and expressed as one hypothetical tank. The H08 model assumes that the runoff generated in a land grid-cell first runs into this tank and excess water flows into a river. If

the water requirement is still not met, water is optionally abstracted from a purely hypothetical unlimited source termed NNBW. NNBW implies water abstraction from missing water components, such as nonrenewable (i.e., overexploited) groundwater, water transferred at distance, glacier melt water, and others.

For each source, municipal water is abstracted first, then industrial and irrigation water. Overall, the water abstraction is expressed as follows:



$$Ctot = Criv + Cmsr + Cnnbw \qquad\qquad (A1)$$

where *Ctot* is consumption-based total water abstraction [kg s$^{-1}$].



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

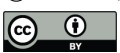



# Tables

Table 1 Geographical data used in this study. Land, river, reservoir, and crop abstraction indicate the land surface hydrology, river routing, crop growth and irrigation water requirement; water abstraction sub-models of H08.

| Data | Source | Sub-model (and newly added schemes) |
|---|---|---|
| Albedo | Dirmeyer et al. (2006) | Land |
| Relief | Harmonized World Soil Database v 1.1 (FAO et al., 2012) | Land, groundwater recharge |
| Soil texture | Soil Texture Map for Global Soil Wetness Project Phase 3 (http://hydro.iis.u-tokyo.ac.jp/~sujan/research/gswp3/soil-texture-map.html) | Land, groundwater recharge |
| Hydrogeology | OneGeology (http://www.onegeology.org/) | Land, groundwater recharge |
| Permafrost and glacier | Circum-Arctic Map of Permafrost and Ground-Ice Conditions (Brown et al., 2002) | Land, groundwater recharge |
| Flow direction map | Döll et al. (2003) | River |
| Aqueduct network | See Table A2 | River, Aqueduct water transfer |
| Reservoirs | Lehner et al. (2011) | Reservoir, Local reservoirs |
| Irrigated area | Siebert et al. (2010) | Crop |
| Crop species | Monfreda et al. (2008) | Crop |
| Crop intensity | Döll and Siebert (2002) | Crop |
| Irrigation efficiency | Döll and Siebert (2002) | Crop, Return flow and delivery loss |
| Municipal water withdrawal | AQUASTAT (www.fao.org/nr/aquastat/) | Abstraction |
| Industrial water withdrawal | AQUASTAT (www.fao.org/nr/aquastat/) | Abstraction |
| Groundwater fraction of water use | Siebert et al. (2010) for irrigation IGRAC (2004) for municipal and industrial | Abstraction, Groundwater abstraction |
| Gross domestic product (GDP) | SSP Database (https://secure.iiasa.ac.at/web-apps/ene/SspDb) | Abstraction, Seawater desalination |



Table 2 Global mean annual fluxes derived from new model components.

| Variable | Source | Methods | Estimation [km³ yr⁻¹] | Note |
|---|---|---|---|---|
| Groundwater recharge | This study | Simulation-based | 13466 | |
| | Döll and Fiedler (2008) | Simulation-based | 12666 | |
| | Wada et al. (2010) | Simulation-based | 15200 | |
| | IGRAC (2004) | Statistics-based | 11795 | |
| Groundwater depletion | This study | Simulation-based | 182 | |
| | Döll et al. (2014) | Simulation-based | 113 | |
| | Wada et al. (2010) | Simulation-based | 283 (±40) | |
| | Pokhrel et al. (2015) | Simulation-based | 330 | |
| | Postel (1999) | Statistics-based | 200 | |
| | Konikow (2011) | Statistics-based | 145 | |
| Aqueduct water transfer | This study | Simulation-based | 199 | |
| Abstraction from local reservoir | This study | Simulation-based | 106 | |
| | Biemans et al. (2011) | Simulation-based | 460 | All (global and local) reservoirs |
| Abstraction from seawater desalination | This study | Simulation-based | 1.8 | |
| | AQUASTAT (FAO, 2016) | Statistics-based | 4.6 | Including non-seawater sources |
| | Hanasaki et al. (2016) | Simulation-based | 2.8 | Base year is 2005 |
| Return flow | This study | Simulation-based | 1546 | |
| | Jägermeyr et al. (2015) | Simulation-based | 1212 | |
| Delivery loss | This study | Simulation-based | 590 | |
| | Jägermeyr et al. (2015) | Simulation-based | 608 | Non-beneficial consumption |





Table 3 Global total groundwater abstraction by sector. Unit: km³yr⁻¹

|  |  | Irrigation | Industrial | Municipal | Total |
|---|---|---|---|---|---|
| This study | Simulation | 551 | 133 | 105 | 789 |
| Siebert et al. (2010) | Simulation | 545 | - | - | - |
| Döll et al. (2014) | Simulation | 490 | 90 | 130 | 710 |
| Wada et al. (2014) | Simulation | - | - | - | 952 |
| Pokhrel et al. (2015) | Simulation | - | - | - | 570 ± 61 |
| IGRAC (2004) | Report | 481 | 150 | 95 | 765* |

* Because some countries do not have a sectorial break down, the total groundwater withdrawal exceeds the sum of the three sectors.





Table 4 River discharge and terrestrial water storage anomaly (TWSA) simulations for heavily human-affected basins. NSE and CC for Nash-Sutcliffe efficiency and correlation coefficient, respectively.

| River | River discharge | | | | TWSA | | | | | | |
|---|---|---|---|---|---|---|---|---|---|---|---|
| | NSE | | Bias | | NSE | | CC | | Slope [mm yr⁻¹] | | |
| | NAT | ALL | NAT | ALL | NAT | ALL | NAT | ALL | NAT | ALL | GRACE |
| Mississippi | 0.67 | 0.71 | 0.08 | 0.00 | 0.34 | 0.24 | 0.60 | 0.57 | -0.66 | -6.78 | 0.50 |
| Parana | -1.87 | -0.73 | 0.28 | 0.24 | 0.25 | 0.54 | 0.80 | 0.88 | -0.63 | -0.47 | -0.28 |
| Chang Jiang | 0.04 | -0.27 | -0.30 | -0.37 | 0.74 | 0.72 | 0.91 | 0.90 | -0.09 | -0.01 | 1.52 |
| Ganges | 0.62 | 0.74 | 0.47 | 0.26 | 0.03 | 0.07 | 0.62 | 0.75 | 3.40 | -21.16 | -10.54 |
| Huang He | 0.14 | -0.02 | 0.05 | -0.29 | -0.27 | 0.18 | 0.52 | 0.67 | 1.87 | -0.97 | -3.77 |
| Colorado | -9.39 | -0.67 | 0.52 | 0.36 | 0.22 | 0.28 | 0.55 | 0.61 | -0.66 | -1.60 | -2.66 |





Table 5 The mean annual volume of water abstraction by sources and sectors. All terms are withdrawal-based, except the results of the original H08 model, which is consumption-based. NNBW is nonlocal and nonrenewable blue water (Hanasaki et al., 2010; Rost et al., 2008).

| | Enhanced H08 (this study) | | | | Original H08 | WaterGAP (Döll et al., 2012) | PCR-GLOBWB (Wada et al., 2014) | AQUASTAT (FAO, 2016) | IGRAC (2004) |
|---|---|---|---|---|---|---|---|---|---|
| | Irrigation | Industrial | Municipal | Total | | | | | |
| River | 1048 | 477 | 260 | 1786 | 541* | - | - | - | - |
| Aqueduct | 182 | 13 | 4 | 199 | - | - | - | - | - |
| Local reservoir | 96 | 10 | 0.4 | 106 | 494* | - | - | - | - |
| Seawater desalination | 0 | 0.4 | 1.4 | 1.8 | - | - | - | - | - |
| Unspecified | 667 | 58 | 21 | 747 | - | - | - | - | - |
| Total surface water | 1993 | 559 | 288 | 2839 | 1035* | 2812 | 3484 | 2911 | - |
| Renewable groundwater | 383 | 125 | 100 | 607 | - | 1271 | 648 | - | - |
| Nonrenewable groundwater | 169 | 8 | 5 | 182 | - | 257 | 304 | - | - |
| Total groundwater | 551 | 133 | 105 | 789 | - | 1528 | 952 | 639 | 765 |
| NNBW | - | - | - | - | 432* | - | - | - | - |
| Total withdrawal | 2544 | 692 | 392 | 3628 | - | 4340 | 4436 | 3550 | - |
| Total consumption | 1368 | 69 | 59 | 1496 | 1466 | 1436 | 1970 | - | - |

*consumptive use.



Table A1 Food and Agriculture Organization of the United Nations (FAO) regions and the country for which complete groundwater information was obtained in IGRAC (2004).

| Region | Nation |
|---|---|
| Eastern Africa | Kenya |
| Middle Africa | Congo |
| Northern Africa | Egypt |
| Southern Africa | South Africa |
| Western Africa | Senegal |
| Northern America | United States of America |
| Central America | Mexico |
| Caribbean | Jamaica |
| South America | Brazil |
| Central Asia | Kazakhstan |
| Eastern Asia | China |
| Southern Asia | India |
| South Eastern Asia | Thailand |
| Western Asia | Turkey |
| Eastern Europe | Hungary |
| Northern Europe | United Kingdom |
| Southern Europe | Italy |
| Western Europe | Germany |
| Australia and New Zealand | Australia |
| Melanesia | Australia |
| Micronesia | Australia |
| Polynesia | Australia |



Table A2 Explicit aqueducts incorporated in this study.

| Regions | Name of aqueduct | Reference |
| --- | --- | --- |
| The Huan He River, China | — | Mingzhou et al. 2007 |
| The Indus River, Pakistan | — | Ullah et al. 2001 |
| The Nile Delta, Egypt | Nasser Canal | NWRP 2005 |
| | Ismaila Canal | |
| | Beheira Canal | |
| | Tawfiki Canal | |
| | Rosetta Branch | |
| | Damietta Branch | |
| | Munufia Canal | |
| The Ganges River, India | Manas-Sankosh-Tista-Ganga Link | NWDA 2016 |
| | Ganga-Damodar-Subernarekha Link | |
| | Subernarekha-Mahanadi Link | |
| The Mahanadi and Godavari Rivers, India | Mahanadi(Manibhadra)-Godavari(Dowlaiswaram) Link | |
| | Godavari(Inchampalli)-Krishna(Nagarjunasagar) Link | |
| | Krishna(Nagarjunasagar)-Pennar(Somasila) Link | |
| | Pennar(Somasila)-Cauvery (Grand Anaicut) Link | |
| The Pennar River, India | Krishna(Srisailam)-Pennar Link | |
| | Krishna(Almatti)-Pennar Link | |
| California, USA | All-American Canal | SWP 2016 |
| | California Aqueduct | |
| | Colorado River Aqueduct | |
| | Friant-Kern Canal | |
| | Hetch Hetchy Aqueduct | |
| | Los Angels Aqueduct | |
| | Western Branch | |
| Israel | National Water Carrier | Feitelson and Rosenthal 2012 |





**Figures**

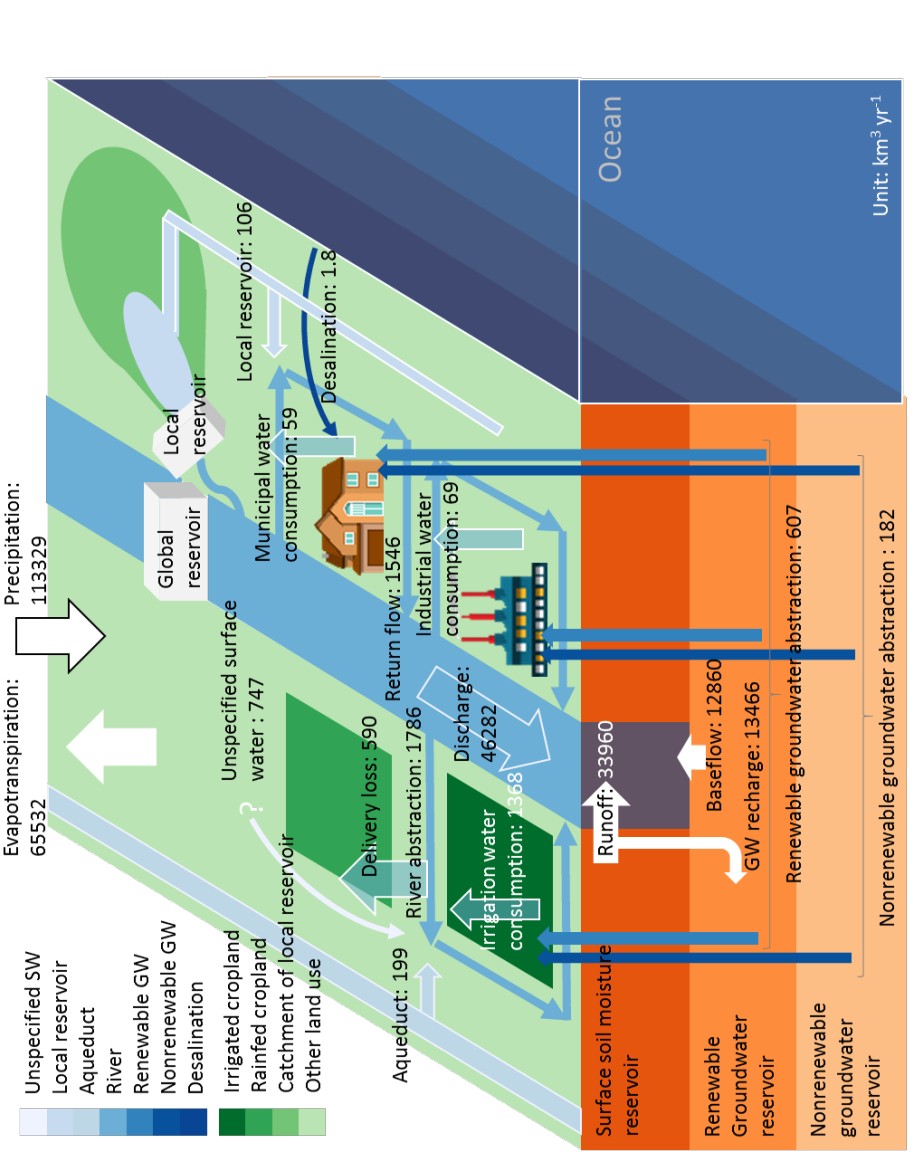

Figure 1 Schematic diagram of the water cycle and abstraction of the enhanced H08. Blue, green, and red symbols denote water, land, and underground reservoirs, respectively. The global reservoirs directly regulate the river flow, while the local reservoirs do not. The numbers are mean annual global water fluxes of key components.



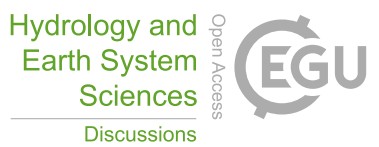

Figure 2 (a) The proportion of the irrigated area equipped with groundwater irrigation (from Siebert et al., 2010). The proportion of the water requirement assigned to groundwater for (b) industrial water, and (c) municipal water (from IGRAC, 2004).

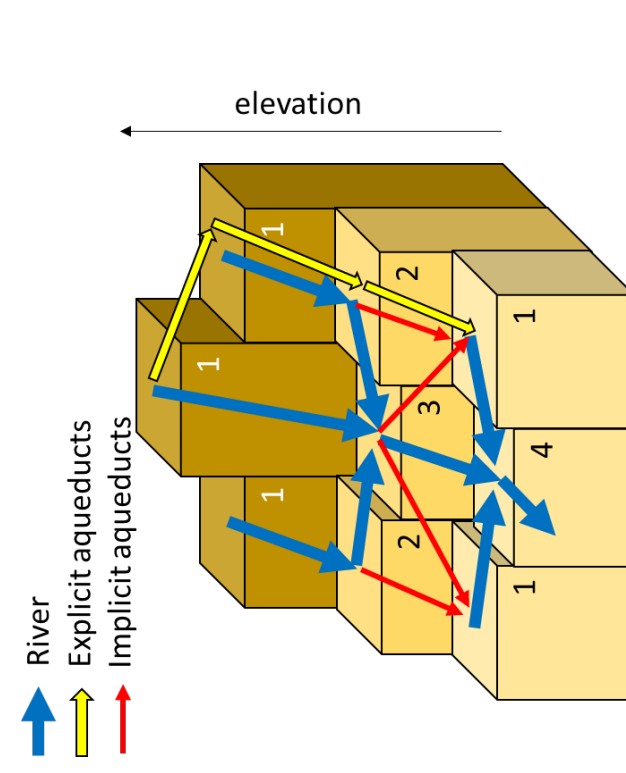

Figure 3 Schematic diagram of aqueduct water transfer for hypothetical 3 × 3 grid cells. Blue arrows delineate the river channel. Yellow and red arrows represent explicit and implicit aquaducts, respectively. The height and number of boxes indicate elevation and the river sequence number. The river sequence increases as the river flows downstream. Water transfer via implicit aqueducts is only assumed to be possible if the elevation and river sequence of the origin are greater than the destinations within the same basin.





Figure 4 River systems of (a) the western USA, (b) central China, (c) the eastern Mediterranean region, and (d) the Indian Subcontinent. Blue lines are the digital river network of DDM30 (Döll et al., 2002). Open inverse triangles are global reservoirs or the reservoirs listed in GRanD (Lehner et al., 2011) with a catchment area larger than 5000 km². Open circles are local reservoirs or the reservoirs with a catchment area not larger than 5000 km². Yellow and red arrows are explicit and implicit aqueducts. For reference, the extent of the Colorado, the Yellow, the Nile, and the Ganges river basins are shown in green.




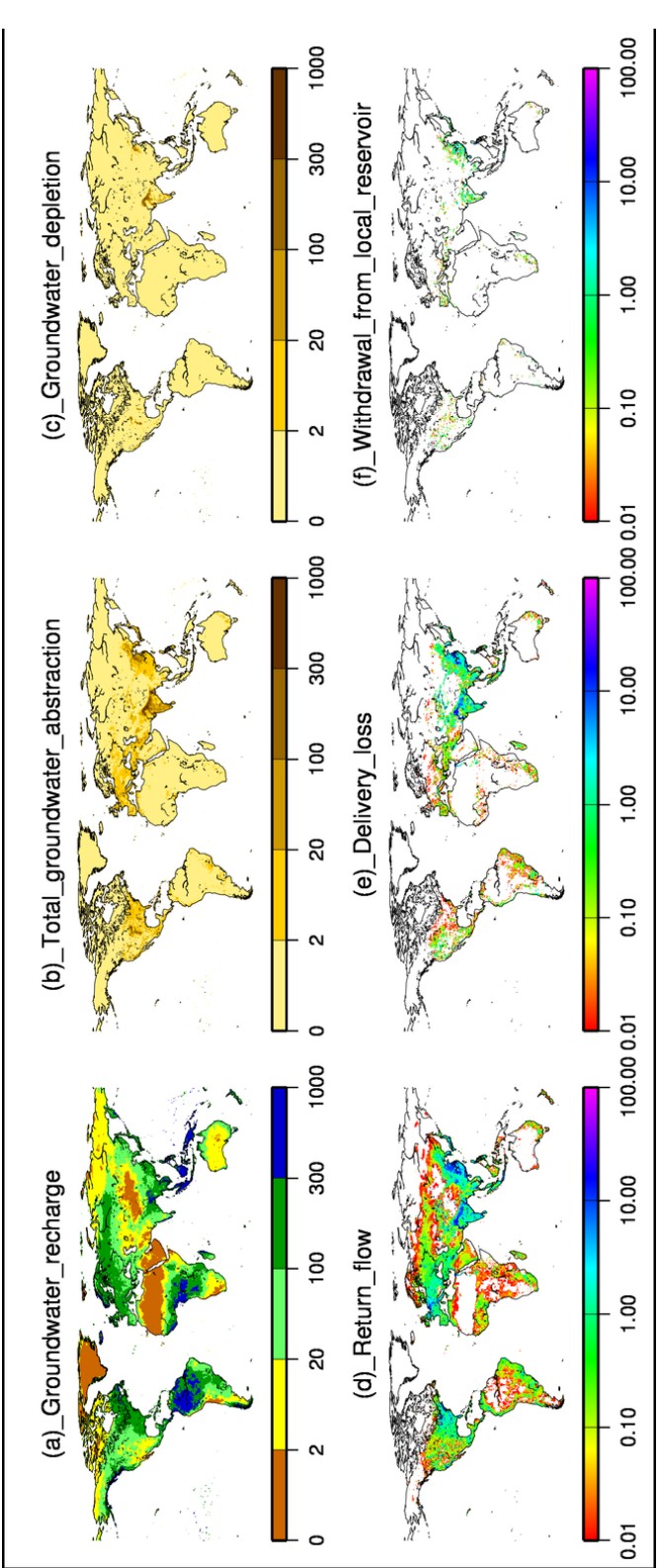

Figure 5 Global distribution of the mean (a) annual groundwater recharge [kg m$^{-2}$ yr$^{-1}$], (b) total groundwater withdrawal [m$^3$ s$^{-1}$], (c) nonrenewable groundwater withdrawal [m$^3$ s$^{-1}$], (d) return flow [m$^3$ s$^{-1}$], (e) delivery loss [m$^3$ s$^{-1}$], and (f) abstraction from local reservoirs [m$^3$ s$^{-1}$].





Figure 6 National estimates of (a) groundwater recharge compared with IGRAC (2004), and
(b) total groundwater abstraction with IGRAC (2004), and
(c) groundwater depletion with Döll et al. (2014). The unit is km$^3$ yr$^{-1}$. The right panels in gray are enlargements of the original figures. Thick and
thin dotted lines show ±20% and +100%/-50% differences, respectively.





Figure 7 Water abstraction from aqueducts of (a) the western USA, (b) central China, (c) the eastern Mediterranean region, and (d) the Indian Subcontinent. The unit is m³s⁻¹. A comparison of the origin and destination of aqueducts is shown in Figure 4.

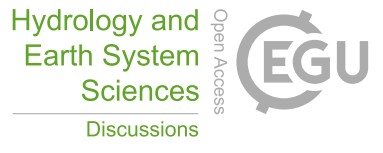



Figure 8 Water abstraction from seawater desalination in (a) the Mediterranean and (b) the Arabian Peninsula. The unit is m$^3$ s$^{-1}$.













Figure 9 Monthly river discharge of (a) the Mississippi River at Vicksburg, (b) the Parana River at Corrientes, (c) the Chang Jiang River at Datong, (d) the Ganges River at Hardinge Bridge, (e) the Huang He River at Huayuankou, and (f) the Colorado River at Lees Ferry.





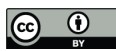


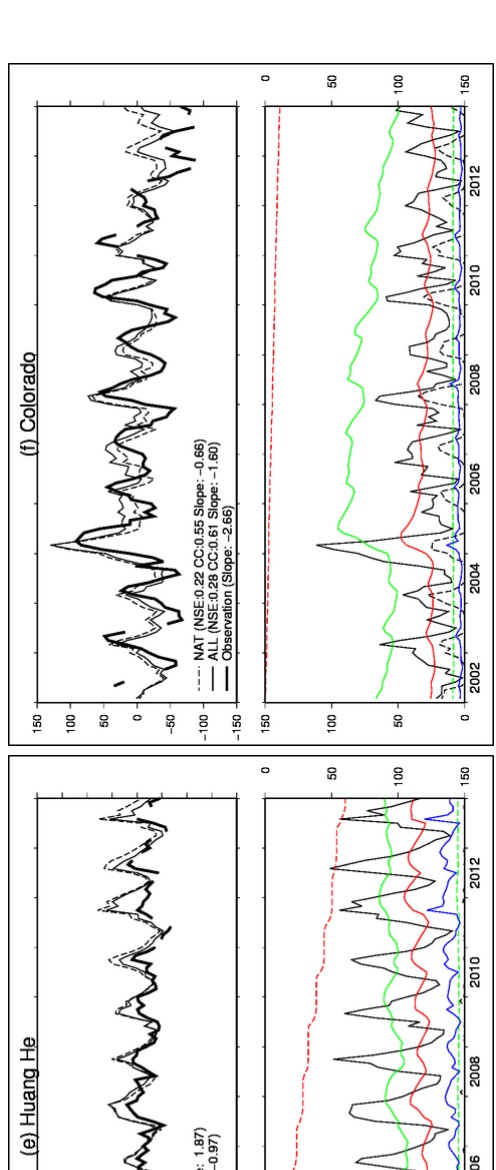

Figure 10 Terrestrial water storage (TWS) of the (a) Mississippi, (b) Parana, (c) Chang Jiang, (d) Ganges, (e) Huang He, and (f) Colorado rivers. The top panel of each figure shows the terrestrial water storage anomaly (TWSA) [mm]. The bottom panel of each figure shows the simulated TWS component [mm]: solid black (soil moisture), broken black (snow water), solid red (renewable groundwater), broken red (storage change in nonrenewable groundwater reservoir, i.e., cumulative volume of nonrenewable groundwater abstraction, right axis), solid green (storage in global reservoirs), broken green (storage in local reservoirs), and solid blue (river water). Note the sign of the cumulative volume of nonrenewable groundwater abstraction, where a positive sign denotes a decrease in water volume.



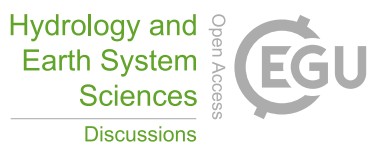
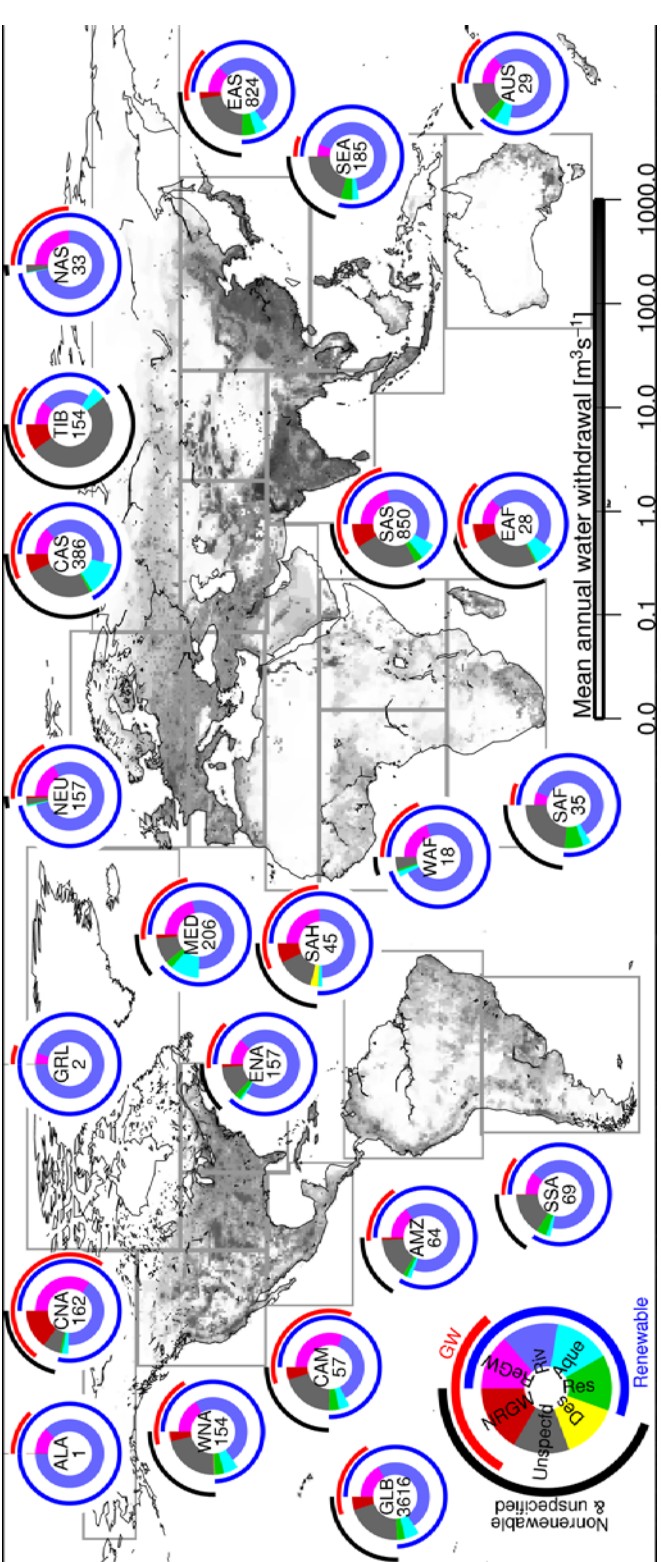

Figure 11 Water sources by region. The global map shows the mean annual total water withdrawal of this simulation. The world is separated into the 21 regions proposed by Giorgi and Francisco (2000). See Table S2 for the full name of the abbreviated regions. For each region, the fractional contribution of each water source is shown in the inner circle: magenta for renewable groundwater, light blue for rivers, cyan for aqueducts, light green for local reservoirs, yellow for seawater desalination, gray for unspecified surface water (USW), and dark red for nonrenewable groundwater. The outer arcs show the aggregated information. Red, blue, and black arcs show the fractional contribution of groundwater, renewable water, and nonrenewable water and USW. The numbers in the circle show the total water withdrawal of the region in km³ yr⁻¹.





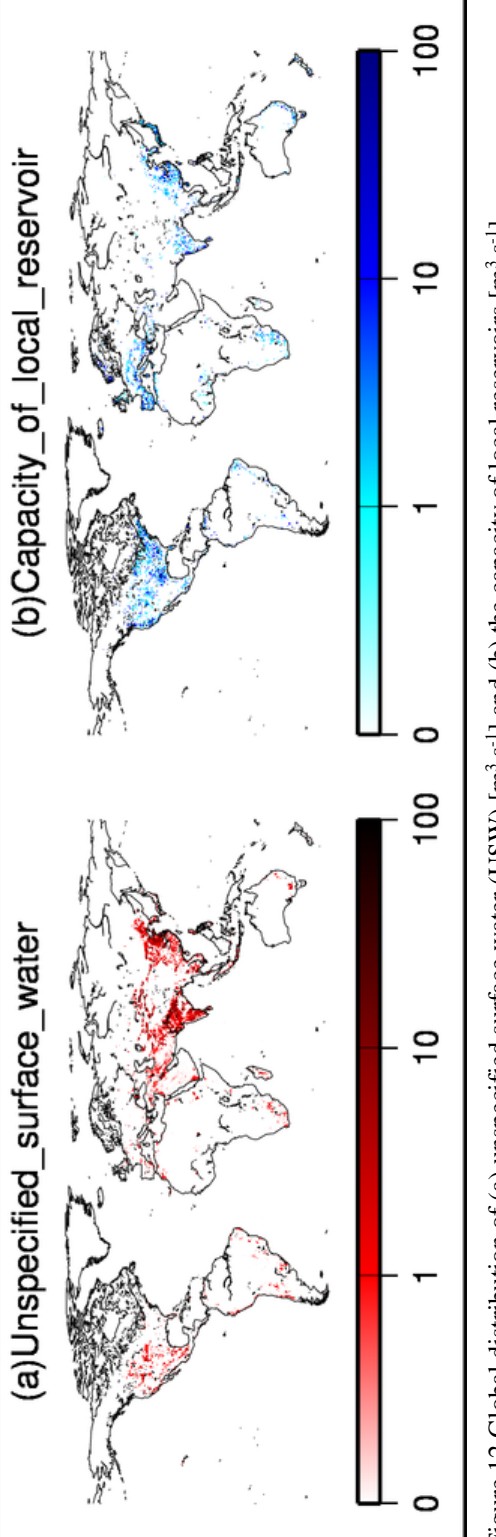

Figure 12 Global distribution of (a) unspecified surface water (USW) [m$^3$ s$^{-1}$] and (b) the capacity of local reservoirs [m$^3$ s$^{-1}$].





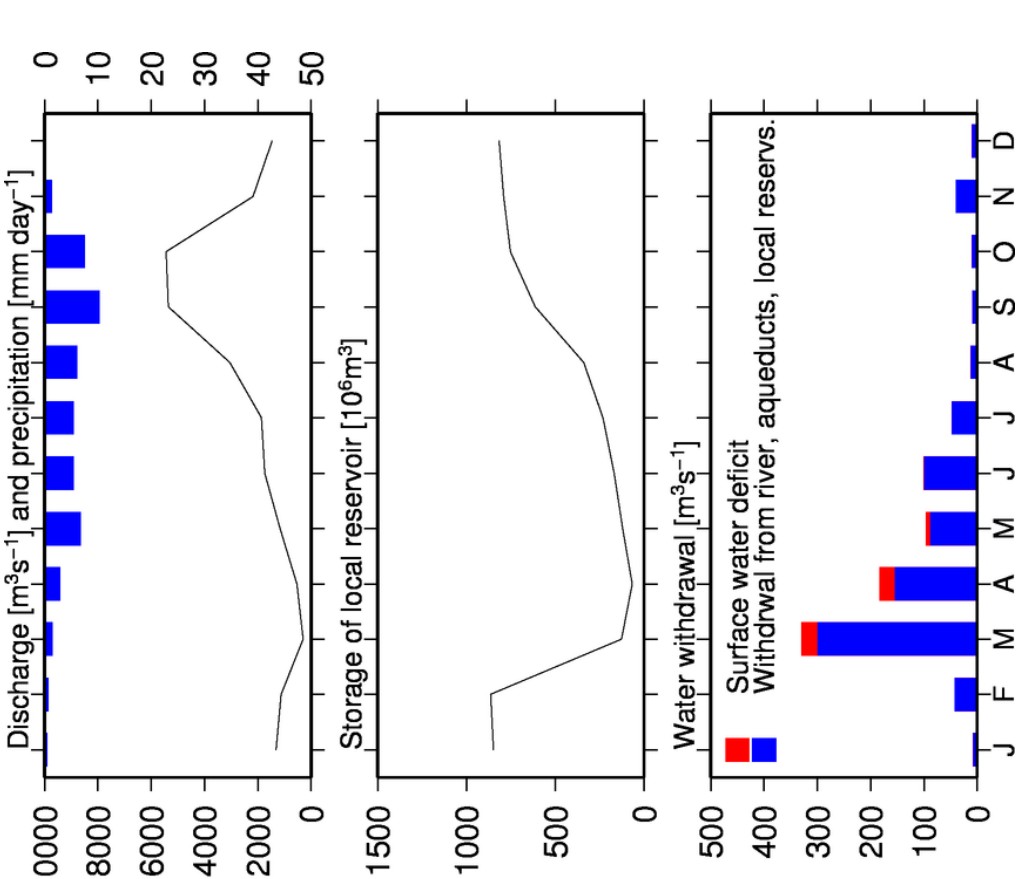

Figure 13 Hydrology and water use in the grid cells near Bangkok, Thailand: (a) mean annual local monthly river discharge [m$^3$s$^{-1}$] and precipitation [mm day$^{-1}$], (b) storage of local reservoirs [10$^6$ m$^3$], and (c) monthly water withdrawal [m$^3$ s$^{-1}$].





Figure A1 Schematic diagram of water abstraction in the original H08 model.