# Peer review of "A global hydrological simulation to specify the sources of water used by humans"

_Hydrology and Earth System Sciences, 2017_

## Referee Comment (RC1) · Anonymous Referee #1 · 23 Jun 2017

**General comments:**

Hanasaki et al present their effort to enhance the H08 global hydrology model with schemes to attribute water abstraction to different water sources. They detail the functionality of every scheme, explain its impacts in different regions and finally discuss some sources of uncertainty that should be kept in mind. The study is clearly structured and (even though it is quite long) easy to read and follow. The combination of this number of water sources in one model definitely merits the publication of their work. However, there are some points I like to be discussed before the final paper should be accepted:

- The authors validate their model version by comparing its results to TWS anomalies measured with GRACE. They utilize a simulation with naturalized setup (e.g. no human impacts) with a simulation with all human impacts. However, the latter does not only include their improvements but also the river regulation and dam management scheme implemented in H08 earlier. I would claim this two aspects are already explain most of the improvement in the TWS anomaly. If the author do want to demonstrate an improvement due to their recent chances, they instead need to compare to a simulation with the original H08 and its human impacts enabled.

- At several points the authors point out that the water balance is strictly closed. Technically this will be true as, no doubt, the models tracks all water storages and fluxes and no water is generated or vanished which the authors are not aware of. However, the do use unlimited water sources to satisfy the water requirements and, thus, the water balance is actually violated. Please reformulate such statements to avoid misunderstandings. The authors present numbers about how much water for a given sector is extracted from which source. However, such number seem to rely on very arbitrary decisions about the order of water extractions (see specific comment P18L4). As the numbers are presented as important parts of their results, I would want to see a justification why this order of water abstraction (and therefore this numbers) is more valid than any other order. Are there economic reasons for this prioritisation? What about allowing the different sectors to share a commonly used source according to their relative water demand fraction. This would much better reflect the simultaneous use of a source by different sectors.

- In section 3.4.3, the authors discuss different reasons about why the available water is significantly less than the required amount of water. Here, I would ask them to reflect about some maybe related points: why do you actually chose to satisfy the water requirements instead of just diagnosing the missing water. In

this way you would also avoid the water balance violation via return flows. based on the information from the appendix I understand that the water withdrawn for industrial and municipal sector is consumed. However, in reality both sectors produce a large amount of waste water which, after treatment, goes back into the water cycle. Is this somehow accounted for or does your data source explicitly include the consumed water? If not this might be part of the missing water. what about the possibility that the water is not actually missing. You derive the surface water / groundwater water withdrawal ratio from quite large scale data. Thus, the real ratio at grid cell level might be extremely different. For grid cells with either a large groundwater or surface water storage the use of this large scale average might cause a depletion in the surface water (groundwater) storage even though there would be enough water in the groundwater (surface water) storage. To me this seems to be a much larger source of uncertainty than e.g. the model resolution itself.

Considering these points, I'd ask the author to either justify the robustness of their existing results or adapt some of my proposed changes where possible. Of course, some points (like missing surface water / groundwater abstraction ratio on grid scale) cannot be changed but should be discussed in the uncertainty part. Alternatively, the authors could consider publishing their research in a journal like GMD (http://www.geoscientific-model-development.net/) where the focus is rather on the development of new model components and, thus, less changes in the manuscript would be needed.

**Specific comments**

- P1L24: Do these numbers refer to the simulation or to the GRACE data?

- P3L14: I am confused about the local reservoirs. So the local reservoirs were
already in the model? It does seem strange to write like ...six things were added, but one not/was already there... please clarify.

- P5L11: How do you know the total water requirement? Is this computed by your model (if so, how) or based on external data (if so, which dataset)? I see it is explained in the appendix, so just add a link here.

- P6L3: While I agree with your decision to use the country with the larger population, I'd like to know what uncertainty is introduced due to it. Would a different sampling affect your fractions distinctively? How important do you consider the (not represented) spatial variation of this fraction within the national borders?

- P6L11: So you fulfil the groundwater abstraction requirements by take water from an unlimited reservoir. Considering that the extracted water will partly end up as irrigation water, some of it will enter the soil and eventually the renewable groundwater storage. How does this agree with your statement in the abstract, that your water balance is closed at any time. For me it sounds like you (at least potentially) add water to the system and therefore effectively violate the water balance (although you probably technically close it by accounting for this violation).

- P7L14: Why is the water transport via aqueducts considered to be a withdrawal. I'd assume you just move water in the river network from one cell to another. Please rephrase.

- P8L11: What is a storage area of a grid cell?

- P8L22: Considering you remark (P3L14) I am now confused about whether this is the old local reservoir scheme of original H08 or the new one that was not implemented...

[Figure]

- P9L12: Does this mean you (simply) define seawater desalination to be equal to the water requirements from municipal and industrial sector? Could you please add an equation as you did for the other sources.

- P9L31: What would water lost through percolation be in you model? I thought you only have one soil layer? What is the storage water percolates from?

- P10L16: I assume you mean you take the water from the origin of an aqueduct that ends in the actual grid cell, right?

- P10L24: Again, using such an unlimited source is a water balance violation.

- P10L26: What do you mean by statistically based OR well validated?

- P11L20: Do you need all of the 8 forcing variables, or just a subset?

- P11L26: From what I read so far, I disagree with this statement. I think you mean that you track all fluxes, sources and sinks and therefore have no unexplained water imbalance in the model, but you can never have a closed water balance while assuming unlimited water reservoirs. Please either reformulated these remarks concerning the water balance or convince me otherwise.

- P11L29: What does it mean with respect to the global reservoirs which are already part of the original H08? Where they active in the NAT simulation as well? Is the difference between NAT and ALL just the use of the new sub-models (thus you can clearly show their effect on the simulation) or is it naturalized vs human-impacted (in which case you would not know whether a given effect comes from the human-impact related processes already being part of the original H08 or from your new processes)? Furthermore, it would be important to know, to what extent unlimited reservoirs contribute to the results.

- P13L17: Or does it rather demonstrate the validity of your irrigation water requirement computation (and not the full model)? Because (as you said yourself) the requirements for other sectors as well as the separation into surface water / groundwater abstraction comes from data.

- P15L22: So also the global reservoirs are only active in the ALL simulation? Which means it is hard to clearly separate the simulation improvement coming from the reservoir operations already being part of H08 and the new source schemes.

- P17L18: Your renewable groundwater storage is rather stable in all six river basins, but the unlimited storage shows a clear trend towards depletion. Looking at equation 4 and 5, I wonder how this can be because I'd expect that first the renewable storage has to run dry (Eq 4) before the unlimited storage is used (Eq 5). Is this an effect of the monthly and/or spatial averaging? Please explain.

- P18L4: Why is this only partly true? If you would withdraw water for irrigation first, probably all water for industrial and municipal sectors would have to come from unsustainable storages. Thus, different numbers for the different sectors do not appear to be results, but rather reflect your computation choice or the amount of water available. As I understand it, the only robust value in those numbers would be the percentage of unsustainable water use (accumulated over all sectors). Please comment on this.

- P19L9: Why 'introducing USW OR taking option 1'? As I understand it, option 1 means introducing the USW.

- P20L2: You mean of volume of the extracted non-renewable groundwater, not the volume of the aquifer itself, right? Please be concise here, because the paragraph sound like the latter.

- P21L17: Is this improvement really due to the six new schemes or rather due to the already existing global reservoirs and dam operations?

- P22L20: I don't see how the economy (maybe apart from desalination part) and environmental aspects are accounted for in H08. From this paragraph I would expect that as a result of H08 simulations you could come up with kind of a cost-benefit analysis for different sources. Thus, this statement seems a bit strong for me. Please be concise about what exactly you can do with this model version.

**Technical comments**

- Tab S1: Please repeat the header for every table page

- Tab A1: It seems this data could be easily displayed in portrait format. Please only use sideways tables if really necessary.

- All figures: You seem to prefer to use landscape format. However, it makes reading the paper more difficult especially in digital format and is not necessary for all of you figures (e.g. 1,4,7,8 and others). Please use portrait whenever possible.

- P8L12: You describe the general mechanics of your scheme. Better use present tense for such paragraphs.

- P8L23: Please revise this paragraph with regard to duplicates (estimation of extent areas) and unnecessary information (implementation difficult, still we implemented it). Your paper is quite long anyway, thus it should be shortened wherever possible.

- Fig 6: In the enlarged figure there is not much to see thanks to the text. As you refer only to a few selected basins in your results section, please remove the labels from most points and add them only to those you actually discuss.

- Fig 8: Consider shading the ocean area in a light grey for an easier overview. Also both regions are so close together that you could show them in one map.

- Fig 11: This is an awesome figure! You may think about changing the color of the region borders to avoid low visibility for patches where the background color matches the border color. Also it might be worthwhile to add small lines from the region to the circle to avoid any confusion about what belonging to what.

- P18L25: Typo CAN → CNA (same typo in table S2)

- P19L8: This is not an inconsistency but just the difference

---

## Referee Comment (RC2) · Anonymous Referee #2 · 12 Jul 2017

GENERAL COMMENTS This manuscript presents an updated version of the H08 GHM that focuses on refining how human water abstractions are modeled at the global scale. Six water sources used for abstraction are focused on here: river flows regulated by large and smaller reservoirs, aqueduct transfers, desalination, renewable and nonrenewable groundwater. Model improvements are largely based on methodologies developed in other studies and results of simulated water fluxes for abstraction are validated against those reported in other peer-reviewed publications. The updated H08 GHM is then used to 1) estimate flows and stocks of natural hydrologic sources and 2) simulate the impact of human water use on natural hydrology both globally and within a subset of major watersheds. This updated model differs from existing GHMs in that no other GHM simultaneously incorporates groundwater recharge, groundwater abstrac-

tion, aqueduct transfers, local reservoirs, desalination and return flow/delivery loss into estimates of global water balances. The work presented here represents an important step forward for GHMs.

SPECIFIC COMMENTS

I am happy to see water infrastructure being more explicitly integrated into GHMs beyond reservoir operations. Aqueducts (Section 2.1.3) and desalination (2.1.5) are important components of human water use that need to be considered as they can have profound impacts on water availability at the regional scale. While I recognize that accounting for these types of infrastructure at the global scale is challenging, it seems that assuming "implicit aqueducts" (e.g., p. 6, lines 23-24) exist to meet water demands may lead to significant overestimation of this form of abstraction, especially given the order of water extraction (e.g., river, global reservoir, aqueduct, local reservoir...). Without any rationale for why this order was selected, I would argue that aqueduct transfers would be far less common than abstractions from local reservoirs. Additional justification on why this particular order was used, or why implicit aqueducts would be very common, would provide needed clarity on this.

What is the benefit of pursuing Option 1 (assuming an imaginary unlimited surface water source) vs. Option 2 (water deficits)? Section 3.4.1 seems to argue that temporal variability does appear in the model and simulates periods where water scarcity exists during which water may be unavailable. From this perspective, it would seem that aligning the model to always have access to an unspecified surface water would diminish this profoundly important problem of scarcity, where deficits are real and serious problems for many, including those irrigating with surface water who may face serious curtailments or crop failures.

Many municipal water systems have significant delivery losses (30-60%), particularly in low-income countries due to a lack of funds for infrastructure repair and deliberate vandalization. Even in the USA, many municipal systems report unaccountedfor water losses of higher than 10%. While I also do not know of any global inventory of water lost during delivery, there are rough estimates available (e.g., http://siteresources.worldbank.org/INTWSS/Resources/WSS8fin4.pdf) that might warrant a re-examination of the assumption that 0.1 and 0.15 (page 10, lines 6-7) are reasonable estimates for this parameter.

TECHNICAL CORRECTIONS

There is a typo on the first line of Section 2.1.7- "fulfil" should be "fulfill"

Figs 5, 6 and 12 are pretty cramped. Finding a way to make these easier to view would be very helpful. (Maybe this won't be an issue if readers can access a high quality version online at publication).

Fig 11 would be even better if there was a nearby or integrated table that reminded readers what each of the three letter codes were. Or, alternately matched pie charts with map areas by a letter (and letters could be tied to region codes in table S2). Right now it's hard to see what matches what section of the map.

---

## Referee Comment (RC3) · Anonymous Referee #3 · 17 Jul 2017

Summary: This Discussion Paper presents on model development in the H08 Global Hydrological Model intended to expand the model's representation of human water appropriation. The reported model enhancements are important in an age of large and increasing human modification of the water cycle, and the authors' approach to model structures and data will be of interest to HESS readership. The Discussion Paper presents both the implementation of model enhancements and the results of model simulations globally and for major river basins. As a presentation of model methods I find the paper to be a strong and useful contribution to the literature—it is clearly written, appropriately detailed, and reports an impressive set of model development and data wrangling efforts. GHM of this type are not my primary research focus, so I cannot comment on the completeness of referencing and the place of this paper in the broader

[Figure]

GHM literature, but it appears that the authors have taken care to put their work in the context of related efforts with different models. As a presentation of simulation results I believe the paper succeeds to some extent. The authors present reasonable estimates of water sources at basin and global scale, and they discuss potential sources of error and uncertainty. Perhaps inevitably, however, both model evaluation and quantification of uncertainty are quite thin. As a result one is left with point estimates of large quantities with significant uncertainties, where it would be considerably more informative to have ranges reported on the basis of some kind of quantitative error estimate. That said, I believe that the Discussion Paper is important in that it presents on a large investment in model development for H08, and my comments below are suggestions for improvements rather than requirements for final publication.

Suggestions:

1. P. 4, line 31-2: This is the first instance I noted in the paper of parameters presented as single point estimates with no sensitivity test and little in the way of justification. The same happens at several otehr points in the text, either explicitly or implicitly via citations. This left me wondering how sensitive model results are to the choice of parameters that are, at best, only roughly constrained by data. Later in the paper the authors identify a number of major sources of uncertainty, but parameters within the hydrological model or management modules are not specifically discussed. Would it be possible to perform targeted sensitivity tests of some of these parameters? Not only would this enhance confidence in model results, but it would provide useful guidance for later model development regarding the relative importance of various parameters to model results.

2. p.5, line 27-28: The assumption regarding the division between surface water and groundwater is quite a large assumption, and it doesn't account for regions in which farmers use groundwater to make up for surface water shortfalls. I don't have any better idea about how to deal with these complications in a global simulation, but it would be useful if the authors could spend a sentence or two justifying the assumption

and explaining potential limitations.

3. Table 4 presents some model evaluation, but no significance tests are presented to show whether ALL is significantly different from NAT for each basin, or whether either simulation is significantly different from observation. Please provide tests of significance for these differences, accounting for temporal autocorrelation as appropriate.

4. Irrigated area: Perhaps this is covered in an earlier H08 publication, but how does the model decide on what fraction of area equipped for irrigation is active in any given year? In my own work I've found this to be a challenge, particularly when it comes to interannual variability in irrigation demand under extended drought—e.g., when farmers fallow irrigation fields due to water shortage. Is this addressed in the model, particularly when it comes to trends in water stressed regions?

5. Comparisons with GRACE: the authors have compared to a single GRACE product. While I expect that different flavors of the spherical harmonics GRACE simulations will be similar in most basins, the more recent mascon solutions have emerged as likely more reliable for terrestrial applications (http://onlinelibrary.wiley.com/doi/10.1002/2016WR019494/full). The authors should consider adding a mascon analysis to their evaluation, both to quantify observation-based uncertainty and because the mascons might indicate that the TWS trends are actually larger than the spherical harmonics solutions indicate, and are in better agreement with ALL simulation results.

Minor comments:

Abstract: The "R" in GRACE stands for Recovery, not Retrieval.

Section 2.2.2: A few words on the WATCH methodology would be helpful for those of us not familiar with it.

Section 3.2.2: Were scaling factors applied to the GRACE data?

---

## Author Comment (AC1) · 6 Sep 2017

General comments:

Hanasaki et al present their effort to enhance the H08 global hydrology model with schemes to attribute water abstraction to different water sources. They detail the functionality of every scheme, explain its impacts in different regions and finally discuss some sources of uncertainty that should be kept in mind. The study is clearly structured and (even though it is quite long) easy to read and follow. The combination of this number of water sources in one model definitely merits the publication of their work. However, there are some points I like to be discussed before the final paper should be accepted:

> Thank you very much for taking the time to review this paper. We are grateful for your detailed and helpful comments. We have responded to all comments and made revisions as indicated below.

• [R1-M1] The authors validate their model version by comparing its results to TWS anomalies measured with GRACE. They utilize a simulation with naturalized setup (e.g. no human impacts) with a simulation with all human impacts. However, the latter does not only include their improvements but also the river regulation and dam management scheme implemented in H08 earlier. I would claim this two aspects are already explain most of the improvement in the TWS anomaly. If the author do want to demonstrate an improvement due to their recent chances, they instead need to compare to a simulation with the original H08 and its human impacts enabled.

> As you have pointed out, we compared naturalized and human-impacted simulations, not the new and old versions of H08. We have rephrased the text in the revised manuscript to make this clear. Following your suggestion, we have added a new simulation mode to run the new H08 model using a configuration similar to that of the original model (we termed this the ORIG simulation). We have described the method and the difference in simulation performance between ALL and ORIG in Supplemental Text S4 as follows:
>
> "The H08 model has been enhanced by six new schemes. We disabled some of its components, such that it works similarly to the original H08 (hereafter the ORIG simulation mode). To disable the groundwater scheme, we set the groundwater recharge factors (i.e., $f_r$, $f_t$, $f_h$, and $f_{pg}$ in Eq. 1) to zero globally. Thus, groundwater recharge is disabled, and groundwater fluxes and storage become constant at zero.

To disable the groundwater abstraction scheme, we set the fraction of the water requirement assigned to groundwater ($f_{gw}$ in Eqs.4 and 5) to zero globally. This setting assigns the entire water requirement to surface water, preventing water abstraction from non-renewable groundwater. To disable aqueduct water transfer and seawater desalination, we set empty maps of implicit and explicit aqueducts, and the area utilizing seawater desalination. To disable return flow and delivery loss, we set the ratio of consumption to withdrawal (e in Eq. 10) and the proportion lost during delivery (l in Eq. 11) to unity and zero globally, respectively. We then fed the consumption-based (not withdrawal-based, as in the main text) water requirement into the H08 model. Finally, to reconfigure the original local reservoirs, we set the catchment area of a local reservoir ($A_{lres}$ in Eq. 7) to unity globally, and then fed the original global and local reservoir distributions into the model.

We compared performance metrics of ORIG with ALL (H08 with new schemes) for the heavily human-affected basins described in Table S4. Regarding TWSA, ALL outperformed ORIG in five of six basins in terms of NSE and CC. The good performance of ALL in the TWS anomaly is attributable primarily to the inclusion of the groundwater recharge scheme, which provides greater amplitude and a delayed peak in the TWS anomaly, agreeing well with observations. Other factors, e.g., the inclusion of return flow and aqueduct water transfer, showed marginal effects because they have little effect on monthly-scale water storage in the basins. Regarding river discharge, we observed considerable improvement in NSE in four of six basins. This result is attributed to the inclusion of groundwater, which supplies stable baseflow throughout the year."

• [R1-M2] At several points the authors point out that the water balance is strictly closed. Technically this will be true as, no doubt, the models tracks all water storages and fluxes and no water is generated or vanished which the authors are not aware of. However, the do use unlimited water sources to satisfy the water requirements and, thus, the water balance is actually violated. Please reformulate such statements to avoid misunderstandings.

To make our intentions clearer, we have rephrased this as "water source is traceable" throughout the text.

[R1-M3] The authors present numbers about how much water for a given sector is extracted from which source. However, such number seem to rely on very arbitrary decisions about the

order of water extractions (see specific comment P18L4). As the numbers are presented as important parts of their results, I would want to see a justification why this order of water abstraction (and therefore this numbers) is more valid than any other order. Are there economic reasons for this prioritisation? What about allowing the different sectors to share a commonly used source according to their relative water demand fraction. This would much better reflect the simultaneous use of a source by different sectors.

> As you have pointed out, the order of water abstractions affects the results. We added the rationale for our priority assessments to Section 2.1.7., as follows: "The order of water withdrawal reflects the distinct differences in water use intensity on the general premise that priority should be given to high value–added products in resource allocation. Municipal, industrial, and agricultural water use intensities per value added (service, manufacturing and power generation, and agricultural sectors) are estimated to be 0.012, 0.063, and 2.2 $10^6 m^3$ $10^6 USD^{-1}$, respectively." The idea of allocating water by the volume of water required is interesting, and perhaps this approach is in practice in some regions. However, this method would not likely yield practical results if it were adopted as a globally uniform algorithm. For example, in Asian countries, irrigation water accounts for a considerable fraction of water used; hence, if the algorithm were adopted, only a small volume of water would be assigned to municipal and industrial sectors during the dry season, which is unrealistic.

• [R1-M4] In section 3.4.3, the authors discuss different reasons about why the available water is significantly less than the required amount of water. Here, I would ask them to reflect about some maybe related points: why do you actually chose to satisfy the water requirements instead of just diagnosing the missing water. In this way you would also avoid the water balance violation via return flows.

> Thank you for this interesting question. We had logical and technical reasons for making this choice. To explain the logical reason, we added the following text to Sect 2.1.7: "A precondition of this study was that water withdrawal estimates were based on values reported in the AQUASTAT database. As shown in Section 2.2.1 and Appendix A1, municipal and industrial water requirements were taken from the database, and simulated irrigation water was carefully compared with these data (Supplemental Text S1). We presumed that AQUASTAT reported the volume of water that was actually abstracted in each nation. Option 1 strictly followed this condition and compensated with unspecified surface water in cases where surface

water data were not available." Regarding the technical reason, we added the following explanation: "Note that adding water from imaginary sources to the H08 model is nearly the only way to quantitatively estimate the volume of missing source water, particularly for irrigation. As shown in Appendix A1, the irrigation water requirement was determined by the soil moisture deficit, which shows highly nonlinear behavior and interacts with other components. An imaginary source of water fills the deficit at every calculation interval, such that the accumulation of water equals the volume of missing source water."

[R1-M5] based on the information from the appendix I understand that the water withdrawn for industrial and municipal sector is consumed. However, in reality both sectors produce a large amount of waste water which, after treatment, goes back into the water cycle. Is this somehow accounted for or does your data source explicitly include the consumed water?

Yes, this water is accounted for. Fractions of the water withdrawn as municipal and industrial water (15% and 10%, respectively) are consumed and removed from the system. The remaining fraction is drained to rivers as return flow. See the latter part of Section 2.1.6.

[R1-M6] If not this might be part of the missing water. what about the possibility that the water is not actually missing. You derive the surface water / groundwater water withdrawal ratio from quite large scale data. Thus, the real ratio at grid cell level might be extremely different. For grid cells with either a large groundwater or surface water storage the use of this large scale average might cause a depletion in the surface water (groundwater) storage even though there would be enough water in the groundwater (surface water) storage. To me this seems to be a much larger source of uncertainty than e.g. the model resolution itself.

Yes, a substantial volume of unspecified surface water could be attributed to the fixed surface and groundwater fraction. We added this point to Section 3.4.3 (Potential sources of uncertainty) as "Water source separation into surface water and groundwater was determined by a single factor, termed the fraction of water requirement allocated to groundwater. Due to a lack of available data, the same factor was applied for vast areas, ignoring local heterogeneity, which is also a source of uncertainty." Additionally, as mentioned in our response to a comment from Reviewer 3, we conducted a new simulation (SWT) that allows additional abstraction from renewable groundwater in cases where unspecified surface water is used. The

results and discussion of this simulation are shown in Supplemental Text S5. In short, the option reduced the volume of unspecified surface water by approximately 200 $km^3yr^{-1}$ (30%), but increased the total groundwater use far in excess of the reported estimation range; hence, this option is less likely to improve the overall simulation performance.

[R1-M7] Considering these points, I'd ask the author to either justify the robustness of their existing results or adapt some of my proposed changes where possible. Of course, some points (like missing surface water / groundwater abstraction ratio on grid scale) cannot be changed but should be discussed in the uncertainty part. Alternatively, the authors could consider publishing their research in a journal like GMD (http://www.geoscientificmodel-development.net/) where the focus is rather on the development of new model components and, thus, less changes in the manuscript would be needed.

> As stated above, we tried to incorporate your valuable comments as frequently as possible. We believe that these additions further enhance the robustness of this paper.

Specific comments

• [R1-S1] P1L24: Do these numbers refer to the simulation or to the GRACE data?

> These numbers refer to the simulation. We have added "simulated" in the text to improve clarity.

• [R1-S2] P3L14: I am confused about the local reservoirs. So the local reservoirs were already in the model? It does seem strange to write like ...six things were added, but one not/was already there... please clarify.

> We have rephrased this section as "Six schemes or additional components were developed and implemented in the H08 model (Hanasaki et al. 2008a, b, 2010, 2013a, b): groundwater recharge, groundwater abstraction, aqueduct water transfer, local reservoirs, seawater desalination, and return flow and delivery loss schemes. Note that the local reservoir scheme was replaced with that of the original H08 model, whereas the other five schemes were new additions."

• [R1-S3] P5L11: How do you know the total water requirement? Is this computed by your model (if so, how) or based on external data (if so, which dataset)? I see it is explained in the appendix, so just add a link here.

Thank you for the suggestion; we have added the link to Appendix A.

• [R1-S4] P6L3: While I agree with your decision to use the country with the larger population, I'd like to know what uncertainty is introduced due to it. Would a different sampling affect your fractions distinctively? How important do you consider the (not represented) spatial variation of this fraction within the national borders?

We have added the following discussion: "As the groundwater use fraction varies considerably among countries (and among regions within countries), this assumption propagates notable uncertainties in the results." We speculate that considerable spatial variation must exist within each nation, but making a more specific statement on this subject is difficult due to a lack of data.

• [R1-S5] P6L11: So you fulfil the groundwater abstraction requirements by take water from an unlimited reservoir. Considering that the extracted water will partly end up as irrigation water, some of it will enter the soil and eventually the renewable groundwater storage. How does this agree with your statement in the abstract, that your water balance is closed at any time. For me it sounds like you (at least potentially) add water to the system and therefore effectively violate the water balance (although you probably technically close it by accounting for this violation).

Your point is well taken. As mentioned above, we rephrased this text in the Abstract, from "the water balance was always strictly closed" to "all water fluxes and storage were strictly traceable." We hope that this adjustment resolves the conflict between the model concept and the wording of the text in the abstract.

• [R1-S6] P7L14: Why is the water transport via aqueducts considered to be a withdrawal. I'd assume you just move water in the river network from one cell to another. Please rephrase.

The primary objective of this paper was to specify water sources for human use in a model simulation. The quantity and location of the movement of "water in the river network from one cell to another" are important in meeting this objective. We have

added the caveat, "Note that water withdrawal via aqueducts is generally not distinguished from water withdrawal from a river in reality, and is seldom recorded independently. This point is revisited in Section 3.1.3."

• [R1-S7] P8L11: What is a storage area of a grid cell?

Thank you for pointing out this oversight. We have rephrased this text as "The catchment area of a local reservoir was equal to that of the largest within a grid cell, unless the area did not exceed the area of the grid cell."

• [R1-S8] P8L22: Considering you remark (P3L14) I am now confused about whether this is the old local reservoir scheme of original H08 or the new one that was not implemented...

In this section, we described the new local reservoir scheme, which was replaced with the old scheme from the original H08 model. The treatment differs substantially from the original, which is shown in Appendix A. Because we have already rephrased the earlier remark (P3L14), this part should now follow logically.

• [R1-S9] P9L12: Does this mean you (simply) define seawater desalination to be equal to the water requirements from municipal and industrial sector? Could you please add an equation as you did for the other sources.

Yes, you are correct. As the condition is complex (i.e., seawater is available for municipal and industrial water use only where the three geographical conditions are met), the mathematical equation is so complex as to be unhelpful for the reader. We would prefer to leave this equation out of the manuscript.

• [R1-S10] P9L31: What would water lost through percolation be in you model? I thought you only have one soil layer? What is the storage water percolates from?

Our original meaning was that "water consumption" due to leakage and similar effects was added to the return flow. As this section appears to be confusing, we have removed this sentence from the revised manuscript.

• [R1-S11] P10L16: I assume you mean you take the water from the origin of an aqueduct that ends in the actual grid cell, right?

Yes, you are correct. Although this sentence is long, we believe that it conveys our intention to readers.

• [R1-S12] P10L24: Again, using such an unlimited source is a water balance violation.

We have rephrased "maintaining the water balance" as "strictly tracking all water fluxes and storage; therefore, the model contains no unexplained water imbalance."

• [R1-S13] P10L26: What do you mean by statistically based OR well validated?

We have rephrased this section as follows: "statistically based (i.e., the national annual water withdrawal volume for municipal and industrial use was derived from the AQUASTAT database; Appendix A) or well validated (i.e., the national annual simulated irrigation water withdrawal volume agrees well with AQUASTAT data; Appendix A, Figure S2)."

• [R1-S14] P11L20: Do you need all of the 8 forcing variables, or just a subset?

We used all eight forcing variables, which are needed to solve the surface energy balance. We added the following sentence: "All variables are indispensable in the H08 model to solve the land surface water and energy balance."

• [R1-S15] P11L26: From what I read so far, I disagree with this statement. I think you mean that you track all fluxes, sources and sinks and therefore have no unexplained water imbalance in the model, but you can never have a closed water balance while assuming unlimited water reservoirs. Please either reformulated these remarks concerning the water balance or convince me otherwise.

We have replaced "strictly maintain the water balance" with "strictly tracking all water fluxes and storage; therefore, the model contains no unexplained water imbalance."

• [R1-S16] P11L29: What does it mean with respect to the global reservoirs which are already part of the original H08? Where they active in the NAT simulation as well? Is the difference between NAT and ALL just the use of the new sub-models (thus you can clearly show their

effect on the simulation) or is it naturalized vs humanimpacted (in which case you would not know whether a given effect comes from the human-impact related processes already being part of the original H08 or from your new processes)? Furthermore, it would be important to know, to what extent unlimited reservoirs contribute to the results.

> To avoid confusion, we have added two sentences: "As mentioned above, the original H08 model consists of six sub-models (land surface hydrology, river routing, reservoir operation, water abstraction, crop growth, and environmental flow requirement). We developed six new schemes. Two simulations with different combinations of sub-models and schemes were conducted in this study." To respond to your questions, global reservoirs were excluded in NAT, which is a naturalized simulation, but included in ALL, which incorporates human impact. Additionally, as mentioned earlier, we added the ORIG simulation (Supplemental Text S4) to reproduce the H08 model simulation with the original settings. Unlimited water sources can appear in the ALL and ORIG simulations when the water abstraction sub-model is enabled.

• [R1-S17] P13L17: Or does it rather demonstrate the validity of your irrigation water requirement computation (and not the full model)? Because (as you said yourself) the requirements for other sectors as well as the separation into surface water / groundwater abstraction comes from data.

> To avoid confusion, we have replaced "implies the validity of our model" with "demonstrates the validity of the irrigation water requirement computation."

• [R1-S18] P15L22: So also the global reservoirs are only active in the ALL simulation? Which means it is hard to clearly separate the simulation improvement coming from the reservoir operations already being part of H08 and the new source schemes.

> Global reservoirs are active in the ALL simulation, but not in the NAT simulation. We also performed the ORIG simulation, which reproduces the function and configuration of the original H08 model. Differences between ALL and ORIG show the effects of the new schemes, as described in Supplemental Text S4. We added the following sentence to the end of introductory paragraph of Section 3.2: "In this subsection, we compare the NAT and ALL simulations to investigate their performance in representing human activity in the enhanced H08 model. A direct

comparison between the original and enhanced H08 models is shown in Supplemental Text S4."

• [R1-S19] P17L18: Your renewable groundwater storage is rather stable in all six river basins, but the unlimited storage shows a clear trend towards depletion. Looking at equation 4 and 5, I wonder how this can be because I'd expect that first the renewable storage has to run dry (Eq 4) before the unlimited storage is used (Eq 5). Is this an effect of the monthly and/or spatial averaging? Please explain.

> Thank you for bringing this to our attention. We have added the following description: "Although non-renewable groundwater storage shows a negative trend, basin-average renewable groundwater storage (Figure 10) was not necessarily depleted because abstraction from the non-renewable part took place only in a limited number of grid cells."

• [R1-S20] P18L4: Why is this only partly true? If you would withdraw water for irrigation first, probably all water for industrial and municipal sectors would have to come from unsustainable storages. Thus, different numbers for the different sectors do not appear to be results, but rather reflect your computation choice or the amount of water available. As I understand it, the only robust value in those numbers would be the percentage of unsustainable water use (accumulated over all sectors). Please comment on this.

> I agree with your assessment that the fraction of non-renewable groundwater was determined by the sectoral order of abstraction (highest priority given to the municipality, followed by industry and irrigation). We have removed "partly" from the text. As we stated above, however, we retained the original order because we believe that it represents general water use priority in the real world.

• [R1-S21] P19L9: Why 'introducing USW OR taking option 1'? As I understand it, option 1 means introducing the USW.

> Thank you for pointing out our error. The text now reads, "by introducing USW (see description of Option 1 in Section 2.1.7)."

• [R1-S22] P20L2: You mean of volume of the extracted non-renewable groundwater, not the volume of the aquifer itself, right? Please be concise here, because the paragraph sound like

the latter.

You are correct. We have carefully edited this section to avoid potential confusion.

• [R1-S23] P21L17: Is this improvement really due to the six new schemes or rather due to the already existing global reservoirs and dam operations?

As stated above, the comparison between ALL and ORIG is shown in Supplemental Text S4. We found that ALL (the H08 model with the new schemes) outperformed ORIG (the original H08 model).

• [R1-S24] P22L20: I don't see how the economy (maybe apart from desalination part) and environmental aspects are accounted for in H08. From this paragraph I would expect that as a result of H08 simulations you could come up with kind of a cost-benefit analysis for different sources. Thus, this statement seems a bit strong for me. Please be concise about what exactly you can do with this model version.

In this section, we intended to emphasize the meaning of specifying water sources and their possible usage in further studies. Indeed, the acquisition of detailed global information about water sources must be the first step toward more advanced studies, including those investigating economic and environmental aspects.

Technical comments

• Tab S1: Please repeat the header for every table page

We will consult the editorial staff about this matter.

• Tab A1: It seems this data could be easily displayed in portrait format. Please only use sideways tables if really necessary.

We wished to avoid frequent page rotation because this often causes editorial problems. Most of the tables fit better in landscape orientation. We apologize for the inconvenience.

• All figures: You seem to prefer to use landscape format. However, it makes reading the paper

more difficult especially in digital format and is not necessary for all of you figures (e.g. 1,4,7,8 and others). Please use portrait whenever possible.

> Again, we apologize for the inconvenience.

• P8L12: You describe the general mechanics of your scheme. Better use present tense for such paragraphs.

> Thank you for the suggestion; we have changed this section to the present tense.

• P8L23: Please revise this paragraph with regard to duplicates (estimation of extent areas) and unnecessary information (implementation difficult, still we implement edit). Your paper is quite long anyway, thus it should be shortened wherever possible.

> We have removed the last sentence of this paragraph, per your suggestion.

• Fig 6: In the enlarged figure there is not much to see thanks to the text. As you refer only to a few selected basins in your results section, please remove the labels from most points and add them only to those you actually discuss.

> Thank you for this suggestion; we have removed as many labels as possible from the enlarged figures.

• Fig 8: Consider shading the ocean area in a light grey for an easier overview. Also both regions are so close together that you could show them in one map.

> We have now shaded the ocean area in gray.

• Fig 11: This is an awesome figure! You may think about changing the color of the region borders to avoid low visibility for patches where the background color matches the border color. Also it might be worthwhile to add small lines from the region to the circle to avoid any confusion about what belonging to what.

> Thank you for this comment. We have changed the color of the regional frame to blue, per your suggestion. In consideration of your comments and those of Reviewer 2, we have labeled each region to avoid potential confusion.

• P18L25: Typo CAN→CNA (same typo in table S2)

> Thank you; this error has been corrected.

• P19L8: This is not an inconsistency but just the difference

> We prefer the term "inconsistency" because water supply and demand are always matched in reality (as in economic theory). From this perspective, water availability and requirement should be consistent.

---

## Author Comment (AC2) · 6 Sep 2017

GENERAL COMMENTS

This manuscript presents an updated version of the H08 GHM that focuses on refining how human water abstractions are modeled at the global scale. Six water sources used for abstraction are focused on here: river flows regulated by large and smaller reservoirs, aqueduct transfers, desalination, renewable and nonrenewable groundwater. Model improvements are largely based on methodologies developed in other studies and results of simulated water fluxes for abstraction are validated against those reported in other peer-reviewed publications. The updated H08 GHM is then used to 1) estimate flows and stocks of natural hydrologic sources and 2) simulate the impact of human water use on natural hydrology both globally and within a subset of major watersheds. This updated model differs from existing GHMs in that no other GHM simultaneously incorporates groundwater recharge, groundwater abstraction, aqueduct transfers, local reservoirs, desalination and return flow/delivery loss into estimates of global water balances. The work presented here represents an important step forward for GHMs.

> Thank you for summarizing the key significance of our work. We appreciate your taking the time to review this paper.

SPECIFIC COMMENTS

[R2-M1] I am happy to see water infrastructure being more explicitly integrated into GHMs beyond reservoir operations. Aqueducts (Section 2.1.3) and desalination (2.1.5) are important components of human water use that need to be considered as they can have profound impacts on water availability at the regional scale. While I recognize that accounting for these types of infrastructure at the global scale is challenging, it seems that assuming "implicit aqueducts" (e.g.,p. 6,lines23-24) exist to meet water demands may lead to significant overestimation of this form of abstraction, especially given the order of water extraction (e.g., river, global reservoir, aqueduct, local reservoir…).

> We have added the following explanation of implicit aqueducts to Section 2.1.3: "As most global hydrological models are grid based, water source is restricted within a grid cell unless aqueducts are present. This condition may result in the production of an artificial gap in water availability in a single basin (i.e., rich in cells with main river channels and poor in neighboring cells without). Implicit aqueducts express the diversion of water in major rivers to surrounding grid cells, reflecting our general

observation that river water is well transferred within a basin, particularly in major river basins in temperate zones. Hence, water availability seldom differs drastically with distance from main river channels."

[R2-M2] Without any rationale for why this order was selected, I would argue that aqueduct transfers would be far less common than abstractions from local reservoirs. Additional justification on why this particular order was used, or why implicit aqueducts would be very common, would provide needed clarity on this.

> We hope that our previous response also answers this question. Regarding the order, the present algorithm takes water first from the river within a grid cell, then from the major river in the neighboring grid cell, and finally from local reservoirs. For example, downtown Tokyo takes water from two distant rivers (i.e., the second source shown above). Indeed, water abstraction for major cities is sourced from the main stems of distant major rivers that have stable flow throughout the year. We believe that the assumption that some grid cells chronically depend on the river discharge of nearby grid cells is reasonable.

[R2-M3] What is the benefit of pursuing Option 1 (assuming an imaginary unlimited surface water source) vs. Option 2 (water deficits)? Section 3.4.1 seems to argue that temporal variability does appear in the model and simulates periods where water scarcity exists during which water may be unavailable. From this perspective, it would seem that aligning the model to always have access to an unspecified surface water would diminish this profoundly important problem of scarcity, where deficits are real and serious problems for many, including those irrigating with surface water who may face serious curtailments or crop failures.

> Option 1 was needed to keep our simulation aligned with the fundamental precondition of this study, which is that the values reported to the AQUASTAT database are actually withdrawn regularly by every country. The validity of the precondition is not necessarily obvious considering the uncertainties in individual data. Unspecified surface water (USW) was estimated at as much as 700 $km^3$ $yr^{-1}$ globally which is too large to solely attribute it to the lack in performance of H08.
> Option 2 excluded the usage of USW and the volume was turned into water deficit or water scarcity. WeWater deficit is regularly observed in many places of the world, for instance as shown in Fig. 13 in Asian countries in the dry season. Also the global

distribution of USW (Fig. 12) largely overlaps with the reported water stressed regions in some of earlier studies (e.g. Fig 2c of Oki and Kanae, 2006). We speculate that the reality would be in between Options 1 and 2, but making a more specific statement on this subject is difficult due to a lack of data. We revised the related parts in Methods and Results Sections to make our intention clear.

[R2-M4] Many municipal water systems have significant delivery losses (30-60%), particularly in low-income countries due to a lack of funds for infrastructure repair and deliberate vandalization. Even in the USA, many municipal systems report unaccounted for water losses of higher than 10%. While I also do not know of any global inventory of water lost during delivery, there are rough estimates available (e.g., http://siteresources.worldbank.org/INTWSS/Resources/WSS8fin4.pdf) that might warrant a re-examination of the assumption that 0.1 and 0.15 (page 10, lines 6-7) are reasonable estimates for this parameter.

> Thank you for this information. Please note that the water use efficiency that we incorporated in this study (the ratio of water consumption to withdrawal) differs from the water transfer efficiency that you mention (the ratio of water delivered to water users to water dispatched from water suppliers). We agree that these delivery losses could be an important part of the water balance in many regions, however it is not possible to directly include these losses in the current model parameterizations. We were not able to directly include your input in the current version of our model, but we will include it in the next version of our model.

TECHNICAL CORRECTIONS

There is a typo on the first line of Section 2.1.7- "fulfil" should be "fulfill"

> Thank you; we have made this correction.

Figs 5, 6 and 12 are pretty cramped. Finding a way to make these easier to view would be very helpful. (Maybe this won't be an issue if readers can access a high quality version online at publication).

> Thank you. We will try to ensure that quality is maintained during the publication process.

Fig 11 would be even better if there was a nearby or integrated table that reminded readers what each of the three letter codes were. Or, alternately matched pie charts with map areas by a letter (and letters could be tied to region codes in table S2). Right now it's hard to see what matches what section of the map.

Thank you for this suggestion. We have added three-letter regional codes to the map.

---

## Author Comment (AC3) · 6 Sep 2017

Summary: This Discussion Paper presents on model development in the H08 Global Hydrological Model intended to expand the model's representation of human water appropriation. The reported model enhancements are important in an age of large and increasing human modification of the water cycle, and the authors' approach to model structures and data will be of interest to HESS readership. The Discussion Paper presents both the implementation of model enhancements and the results of model simulations globally and for major river basins. As a presentation of model methods I find the paper to be a strong and useful contribution to the literatures. it is clearly written, appropriately detailed, and reports an impressive set of model development and data wrangling efforts. GHM of this type are not my primary research focus, so I cannot comment on the completeness of referencing and the place of this paper in the broader GHM literature, but it appears that the authors have taken care to put their work in the context of related efforts with different models.

We are grateful that you have evaluated our paper so highly.

As a presentation of simulation results I believe the paper succeeds to some extent. The authors present reasonable estimates of water sources at basin and global scale, and they discuss potential sources of error and uncertainty. Perhaps inevitably, however, both model evaluation and quantification of uncertainty are quite thin. As a result one is left with point estimates of large quantities with significant uncertainties, where it would be considerably more informative to have ranges reported on the basis of some kind of quantitative error estimate.

Thank you for this comment. For technical reasons, the addition of systematic/formal error bars to our estimates was quite challenging. We have now added standard deviations to the mean annual estimates, which were derived from 30 years of simulation, for the key simulation results (Tables 2 and 3, Abstract). This addition presents basic information about one aspect of the uncertainty.

That said, I believe that the Discussion Paper is important in that it presents on a large investment in model development for H08, and my comments below are suggestions for improvements rather than requirements for final publication.

Thank you for your valuable comments. All of your points are well taken and we have tried our best to incorporate them in the revised manuscript.

Suggestions:

[R3-M1] P. 4, line 31-2: This is the first instance I noted in the paper of parameters presented as single point estimates with no sensitivity test and little in the way of justification. The same happens at several other points in the text, either explicitly or implicitly via citations. This left me wondering how sensitive model results are to the choice of parameters that are, at best, only roughly constrained by data. Later in the paper the authors identify a number of major sources of uncertainty, but parameters within the hydrological model or management modules are not specifically discussed. Would it be possible to perform targeted sensitivity tests of some of these parameters? Not only would this enhance confidence in model results, but it would provide useful guidance for later model development regarding the relative importance of various parameters to model results.

> We have enhanced the discussion of the hydrological parameter uncertainties in Section 3.4.3 (Potential sources of uncertainty): "Moreover, the hydrological parameters were not tuned to individual basins, yielding a generally lower reproducibility of historical river flow observations (e.g., Hattermann et al. 2017). In cases in which the H08 model was applied to specific basins, sensitivity testing and hydrological parameter calibration were conducted systematically using reliable long-term observations (e.g., Hanasaki et al. 2014; Masood et al. 2014). Conversely, when H08 is applied globally, as in this study, these procedures are difficult to perform because observations are not available for vast areas and simulation periods. Without ground truthing, the sensitivity test cannot be interpreted, and parameter calibration cannot be performed. This is particularly true for groundwater parameters because very few reliable observations representing the grid-cell size (0.5°) are available."

[R3-M2]. p.5, line 27-28: The assumption regarding the division between surface water and groundwater is quite a large assumption, and it doesn't account for regions in which farmers use groundwater to make up for surface water shortfalls. I don't have any better idea about how to deal with these complications in a global simulation, but it would be useful if the authors could spend a sentence or two justifying the assumption and explaining potential limitations.

> We have taken your advice and added a new simulation option that extracts additional renewable groundwater when surface water is depleted. The relevant methods and results are shown in Supplemental Text S5, as follows:

"As described in Sections 2.1.2. and 2.1.7., the water source at an individual grid cell is assigned to the surface water and groundwater parts using the fixed local parameter, termed the fraction of the water requirement assigned to groundwater ($f_{gw}$ in Eqs. 4 and 5). We added a simulation option (hereafter SWT) to abstract additional renewable groundwater when surface water is depleted. This option reflects the ability of some water users to switch water sources by taking availability into account.

The results are shown in Table S4. Compared with the ALL simulation, SWT uses as much as 213 km$^3$ yr$^{-1}$, or approximately 30%, less unspecified surface water. Groundwater abstraction increased by 345 km$^3$ yr$^{-1}$. We used this gap to compensate for the reduction in river water abstraction (123 km$^3$ yr$^{-1}$). Additional groundwater abstraction depressed the storage of renewable groundwater and consequently the baseflow, which eventually reduced the availability of river water. Comparing the total groundwater use of ALL and SWT, the estimation of ALL is closer to the range of statistics-based literature (639–765 km$^3$ yr$^{-1}$, according to FAO 2016 and IGRAC 2004). This result implies that although water users may switch water sources flexibly from surface water to groundwater in some regions, this appears not to be the case in many parts of the world."

[R3-M3]. Table 4 presents some model evaluation, but no significance tests are presented to show whether ALL is significantly different from NAT for each basin, or whether either simulation is significantly different from observation. Please provide tests of significance for these differences, accounting for temporal autocorrelation as appropriate.

We have added statistical testing of the bias, correlation coefficient, and slope results presented in Tables 4 and S3. Please note that we excluded the Nash–Sutcliffe Efficiency because the authors do not feel that there is an established method to conduct statistical significance test of it. In short, we first added statistical significance information to the correlation coefficient and slope (i.e., annual trend) for TWSA. We then added statistical significance testing to the difference between the NAT and ALL simulations for bias in river discharge, correlation coefficient, and slope in TWSA.

[R3-M4]. Irrigated area: Perhaps this is covered in an earlier H08 publication, but how does the model decide on what fraction of area equipped for irrigation is active in any given year?

In my own work I've found this to be a challenge, particularly when it comes to interannual variability in irrigation demand under extended droughts. e.g., when farmers fallow irrigation fields due to water shortage. Is this addressed in the model, particularly when it comes to trends in water stressed regions?

> As described in Section 2.2.1, all land use was fixed throughout the simulation period for this study and the irrigation water requirement was estimated based on this fixed land use. Although interesting to analyze, little geographical information is available on the variability and change in irrigated area (and crop type). We rephrased the text in Section 3.4.3 (Key uncertainties) as: "We note that Siebert et al. (2015) developed the global distribution of irrigated areas from 1900 to 2005, which would be an important contribution to simulations incorporating inter-annual variation in the irrigation water requirement. We fixed the irrigated area throughout the simulation period, however, because little information is available on annual variation in crop practices (e.g., crop type, crop intensity, fractions of surface water and groundwater dependence)."

[R3-M5]. Comparisons with GRACE: the authors have compared to a single GRACE product. While I expect that different flavors of the spherical harmonics GRACE simulations will be similar in most basins, the more recent mascon solutions have emerged as likely more reliable for                                      terrestrial                                      applications (http://onlinelibrary.wiley.com/doi/10.1002/2016WR019494/full). The authors should consider adding a mascon analysis to their evaluation, both to quantify observation based uncertainty and because the mascons might indicate that the TWS trends are actually larger than the spherical harmonics solutions indicate, and are in better agreement with ALL simulation results.

> We obtained the Mascon data (Scanlon et al. 2016) for 12 basins and drew the same figures as Figures 10 and S4. We found only marginal differences between them (Figures R1 and R2 show TWS anomaly in the Mississippi River that adopt CSR and Mascon as observation respectively). The most notable difference was observed in the Ganges River Basin; Mascon showed twice as large a decreasing trend in TWS (-19.59 mm/yr) than in CSR (-10.54 mm/yr), which is consistent with the H08 simulation (-21.16 mm/yr). As no significant difference was observed, we decided to continue using the CSR product.

[Figure]

Figure R1. TWS anomaly in the Mississippi River. Observation adopts CSR.

[Figure]

Figure R2 TWS anomaly in the Mississippi River. Observation adopts Mascon.

Minor comments:

[R3-S1] Abstract: The "R" in GRACE stands for Recovery, not Retrieval.

    Thank you, this has been corrected.

[R3-S2] Section 2.2.2: A few words on the WATCH methodology would be helpful for those of us not familiar with it.

    We have added further description of WFDEI, which reads, "The WATCH forcing methodology represents sub-daily reanalysis data scaled arithmetically to make the mean values and the range of variation consistent with spatio-temporal coarse-ground observation data."

Section 3.2.2: Were scaling factors applied to the GRACE data?

    We applied scaling factors. We now specify this in Section 3.2.2.

---

## Author Response (AR2)

Dear Editor and Reviewers,

We are grateful to you for taking the time to handle and review our paper. Herewith we would like to submit our revised manuscript, entitled "A global hydrological simulation to specify the sources of water used by humans" for consideration for publication in Hydrology and Earth System Sciences. We have made the corrections and modifications suggested by the reviewers as shown below. We hope you will find the modifications are satisfactory and that the manuscript is now suitable for publication in Hydrology and Earth System Sciences. Please let us know if any further clarifications are necessary.

Sincerely yours,
Naota Hanasaki (on behalf of authors)

*Comments from Editor*

*I have received reviews from the two of the original reviewers of this manuscript. Both are favorable reviews. Congratulations! Reviewer #2 provides additional comments for further clarification. I encourage you to please respond to those comments and revise the manuscripts accordingly. Once I receive the revised manuscript I'll send it to the reviewer #2 for a speedy review. Also I'd encourage you to submit revised manuscript with track changes or at least provide the page and line numbers that are revised. This will help expedite the second round of review.*
*Thank you and I look forward to seeing the revised manuscript.*

> Thank you for your positive comments. We have modified the text to fully dispel the concerns of Reviewer 2. We provided page and line numbers that were revised, together with some screenshots showing the revised parts with track changes.

*Reviewer 1*

*I very much thank the authors for taking the time to answering all my questions in details and even doing an additional simulation with their model. For the vast majority of questions, I am very satisfied with the answers. Still, some small issues remain:*

Thank you for taking the time to further investigate our manuscript.

*R1-S2/S8: It might be me, but I still don't get this part. Logically, I assume that your newly developed local reservoir scheme replaced or modified the one already existing in H08. This means your new local reservoir scheme is used, but the one from the original H08 is not (although the new might be based on the old). However, you keep phrasing it the other way around: '… the local reservoir scheme was replaced with that of the original H08 model …' (in contrast to: … the local reservoir scheme replaced that of the original H08 model… ), which implies your newly developed local reservoir scheme is not used in the model.*

Finally we understood the point. You are absolutely correct: now the text is read "the local reservoir scheme was replaced with the new one" (Page 3, lines 14). We are totally sorry that our simple grammatical mistake has bothered you for a long time.

> **2.1. Newly added schemes**
>
> Six schemes or additional components were developed and implemented into H08 (Hanasaki et al. 2008a,b, 2010, 2013a,b), namely, groundwater recharge, groundwater abstraction, aqueduct water transfer, local reservoirs, seawater desalination, and return flow and delivery loss schemes. Note that the local reservoir scheme was replaced with the new one
> 15 , whereas the other five schemes were new additions. Figure 1 shows a schematic diagram of the enhanced H08.
> The description of the individual schemes is provided in the following subsections. Each description begins with a brief

*If this would be really the case, there would be no point in describing the new scheme at all. Sorry for picking at this point, but considering the values in the newly provided table S4, it seems to be important to know exactly which scheme was used.*

We used the new scheme throughout the study. As shown in Supplemental Text S4, exceptionally, the ORIG simulation reproduced the original (i.e. old) scheme.

*As shown by your numbers, for river discharge in heavily human affected basins, even NAT simulations slightly outperform the ORG simulations. Thus, a reader would want to know whether the improvements in ALL result from modifications of the local reservoir scheme itself, by balancing its error with your new, additional schemes or by emergent effects of using all of the schemes together.*

The latter is the case. Let us summarize again the simulation settings (all is noted in body and this is just for your reference).

| | Land surface sub-model | Human sub-models (global/local reservoir operation, water abstraction aqueduct water transfer, desalination) |
|---|---|---|
| ALL | New (including gw) | Including all in new configurations |
| NAT | New (including gw) | Excluding all |
| ORIG | Old (excluding gw) | Including all in old configurations |

The change in performance between ALL and NAT is attributed to inclusion/exclusion of the aggregated effects of all human sub-models. A comparison between NAT and ORIG requires additional care since the land sub-model was also different (i.e. inclusion/exclusion of the groundwater recharge scheme).

*R1-S9: In my view, a complex mathematical relation is rather a reason to publish it than to skip it. I see how you don't want to confuse readers by adding it to the main text, but I would urge you to consider adding it to the appendix or supplements.*

We have newly added the Equation (9) in body (page 9). After consideration, it turned out that the equation can be expressed extremely simple (see the screenshot below for quick reference). In the previous round of revision, we were worrying about how to mathematically express the key conditions (i.e. seawater desalination is only available for regions meeting three economic and geographical conditions; and when it is available, it shuts out water abstraction from other sources), but we finally found that such conditions can be omitted from the equation since they are already written in text. Again, thank you for encouraging us to show in a mathematical form.

15  consecutive 0.5° × 0.5° grid cells (approximately 165 km along the equator) of seashore. By assuming seawater desalination is not used for irrigation, and all of the municipal and industrial water withdrawal in AUSD cells is abstracted by seawater desalination, which is supported by the available statistical records in Hanasaki et al. (2016), we could estimate the quantitative spatiotemporal distribution of withdrawal from seawater desalination. Water withdrawal of seawater desalination ($Wdes$) is expressed as:

20  $$Wdes = \begin{cases} Qreq_{mun} + Qreq_{ind} & (AUSD\ cells) \\ 0 & (non-AUSD\ cells) \end{cases} \quad (9)$$

where $Qreq_{mun}$ and $Qreq_{ind}$ is municipal and industrial water requirement respectively [kg s$^{-1}$].

*Reviewer 2*

*The authors have addressed all of my concerns with the previous version of the manuscript,*

*and I am happy to recommend the paper for final publication.*

> We are grateful for your positive evaluation. We truly appreciate your constructive comments during the review process.